# Shape-memory effects in molecular crystals

Ejaz Ahmed [1,4], Durga Prasad Karothu [1,4], Mark Warren [2] & Panče Naumov[1,3]

Molecular crystals can be bent elastically by expansion or plastically by delamination into slabs that glide along slip planes. Here we report that upon bending, terephthalic acid crystals can undergo a mechanically induced phase transition without delamination and their overall crystal integrity is retained. Such plastically bent crystals act as bimorphs and their phase uniformity can be recovered thermally by taking the crystal over the phase transition temperature. This recovers the original straight shape and the crystal can be bent by a reverse thermal treatment, resulting in shape memory effects akin of those observed with some metal alloys and polymers. We anticipate that similar memory and restorative effects are common for other molecular crystals having metastable polymorphs. The results demonstrate the advantage of using intermolecular interactions to accomplish mechanically adaptive properties with organic solids that bridge the gap between mesophasic and inorganic materials in the materials property space.

[1] New York University Abu Dhabi, PO Box 129188, Abu Dhabi, UAE. [2] Diamond Light Source, Didcot, Oxfordshire OX11 0DE, UK. [3] Radcliffe Institute for Advanced Study, Harvard University, 10 Garden St, Cambridge, MA 02138, USA. [4] These authors contributed equally: Ejaz Ahmed, Durga Prasad Karothu. Correspondence and requests for materials should be addressed to P.N. (email: pance.naumov@nyu.edu)

D ynamic molecular crystals are mechanically compliant materials which combine properties of soft matter such as polymers, and hard matter such as inorganic materials. These materials are thought to hold potentials in fields that range from power microgenerators and smart materials, to artificial muscles and soft robotics. They can effectively transduce energy to work by bending[1–10], jumping[11–19], twisting[20–23], or curling[9,24], and occasionally exhibit properties that are typical for polymers, as it has been recently demonstrated with self-healing organic crystals[4,25–27]. Soft-matter-like properties of molecular crystals can be accomplished by using heat[11,13–15,18,19], light[28–32], and/or pressure[33–39]. The most rapid transduction of energy was observed with thermosalient[4,11–15,18,19,40,41] and photosalient[28–32] crystals; the latent elastic energy that accumulates within these materials upon structural transformation is released instantaneously and results in swift movements, self-propulsion, and even ballistic events[11–16,18,40,41]. However, for applications that require continual operation such as microfluidics or actuating switches, crystals that can deform reversibly and retain their integrity during operation are the preferred material of choice, and this has inspired studies into the mechanistic details and energetic profiles of their deformation[1,20,23,27,29–32]. Although within the realm of classical mechanics bending of a beam-like entity such as a crystal might appear a trivial problem, the analysis, quantification and modeling of reshaping of molecular crystals has proven challenging and is the subject of active research[1,23,29–32].

It is now generally accepted that molecular crystals can be bent either elastically or plastically, depending on their ability to recover their original shape after the action of activating stimulus has been terminated (Supplementary Note 1). Upon elastic and plastic bending of crystals their diffraction ability and single crystal integrity is generally retained, as concluded from their discrete diffraction patterns[8,42–46]. In some cases loss of the discrete diffraction has been observed upon plastic bending[47–49], however, their single crystal nature is retained[50]. The structure determination of bent crystals by using conventional diffraction analysis is not trivial[44] due to the existing range of unit cell orientations and increased defects in the bent section that normally result in streaky diffraction profiles. In effect, although structural mapping in the bent region of an elastically bent crystal was recently reported[8], there is no direct evidence yet of the atomic-scale structure of a plastically deformed crystal. Micro-focus infrared spectroscopy coupled to periodic density functional theory calculations indicated[51] that specific intramolecular vibrations can have predictive power for the structural changes that occur upon bending, however, the combination of expansion, contraction and distortion of the unit cell prevents deconvolution of the deformations that contribute to the spectral profile. Here, with mechanically compliant crystals of terephthalic acid (TA) we report a distinctly different mechanism of plastic bending of molecular single crystals which have multiple phases that can coexist around the temperature at which the deformation is induced, and we provide the crystal structure of the bent section of such plastically bent crystal as direct evidence of the proposed mechanism. We also establish that this plastic deformation which effectively results in coexistence of two phases in the bent section of the crystal is the origin of unconventional properties such as shape-memory and self-restorative effects.

## Results

**Phase transition and shape-restorative effects.** Under specific crystallization conditions (slow evaporation of solutions or hydrothermal crystallization), TA can be obtained as two polymorphs, one of which is stable (form II), while the other is metastable (form I) at ambient conditions[52]. Crystals of both forms are exceptionally robust and can be converted into each other by heating or cooling without visible deterioration (Fig. 1b, Supplementary Fig. 1). The thermal profile of the phase transition confirms that it is of first order[52] and occurs with a thermal hysteresis of about 33 K (Supplementary Fig. 2). The transition occurs by rapid transformation of well-defined parallel domains, resulting in intermediate states where the crystal contains alternating sectors of both phases[4]. These multidomain crystals typically persist for several seconds before the remaining domains are converted to the product phase and the crystal becomes a single phase. The transition of unrestrained crystals is accompanied by sudden strong motion[4] known as thermosalient effect[12,16]. As shown in Fig. 1a, the structures of both forms are layered and composed of parallel molecular tapes aligned in the direction of the crystallographic c axis, and the tapes are slightly offset in the two forms. Upon repeated thermal cycling of the crystal over the phase transition, the molecular layers slide back and forth without other significant structural rearrangements, and the small overall structural change contributes to the retention of crystal integrity. In addition to heating and cooling, the transition can be triggered in both directions by application of local pressure, for instance, by pricking a crystal of the metastable form I obtained by heating and cooling of form II with a sharp, hard object[4,52]. The rapid reshaping upon mechanical stimulation results in the so-called mechanosalient effect, whereby mechanically stimulated crystals spring off from the solid support and move (Fig. 1f; Supplementary Fig. 3 and Supplementary Movies 1–5)[4]. Determination of the unit cell parameters by in situ X-ray diffraction (XRD) and phase identification by Raman spectroscopy before and after the mechanically stimulated phase transition as well as of partially transformed crystals of form I confirmed that during this process they transition to the stable form II (Supplementary Figs. 4 and 5, Supplementary Table 1).

In addition to the thermosalient and mechanosalient phenomena, crystals of this material were found to have unusual shape-memory and shape-restorative properties. Prismatic crystals of form II can be easily bent when localized external pressure is applied on their (010) or (0$\bar{1}$0) faces in a three-point bending geometry, either manually or by using a tensile tester (Fig. 1c, Supplementary Fig. 6, Supplementary Movie 6). However, unlike other examples of plastically bendable crystals[38] the bending of TA crystals occurs without visible delamination. Instead, striations appear on the crystal surface, (001), at an angle to the longest crystal axis. As shown in Fig. 1d and Supplementary Movie 7, when the plastically bent crystals of form II are taken over the phase transition to form I their straight shape is partially recovered, in a manner that resembles the shape-memory effect that is known for some shape-memory polymers and alloys (Supplementary Fig. 8, Supplementary Movies 7–11). For instance, after the transition, mechanically bent crystal of form II with a curvature of 0.51 mm$^{-1}$ typically straightens up to a curvature of 0.18 mm$^{-1}$ (Supplementary Note 3). We noticed that among other factors—such as defects that are beyond the experimenter's control—the ability for recovery of the straight shape is determined by the degree of bending and crystal thickness. In some cases, we were astonished to observe complete recovery of the straight shape of the crystal, based on its curvature (Supplementary Movies 8 and 9). Further, upon cooling to room temperature, form I crystals that have been straightened by heating were transformed to form II, and the initial bent shape of the crystals was recovered. These experiments demonstrate that the shape-memory effect occurs even on cooling; the crystals appear to retain "information" of the deformation in their structure. This transformation between bent and straight shape of the crystal can be repeated several times by

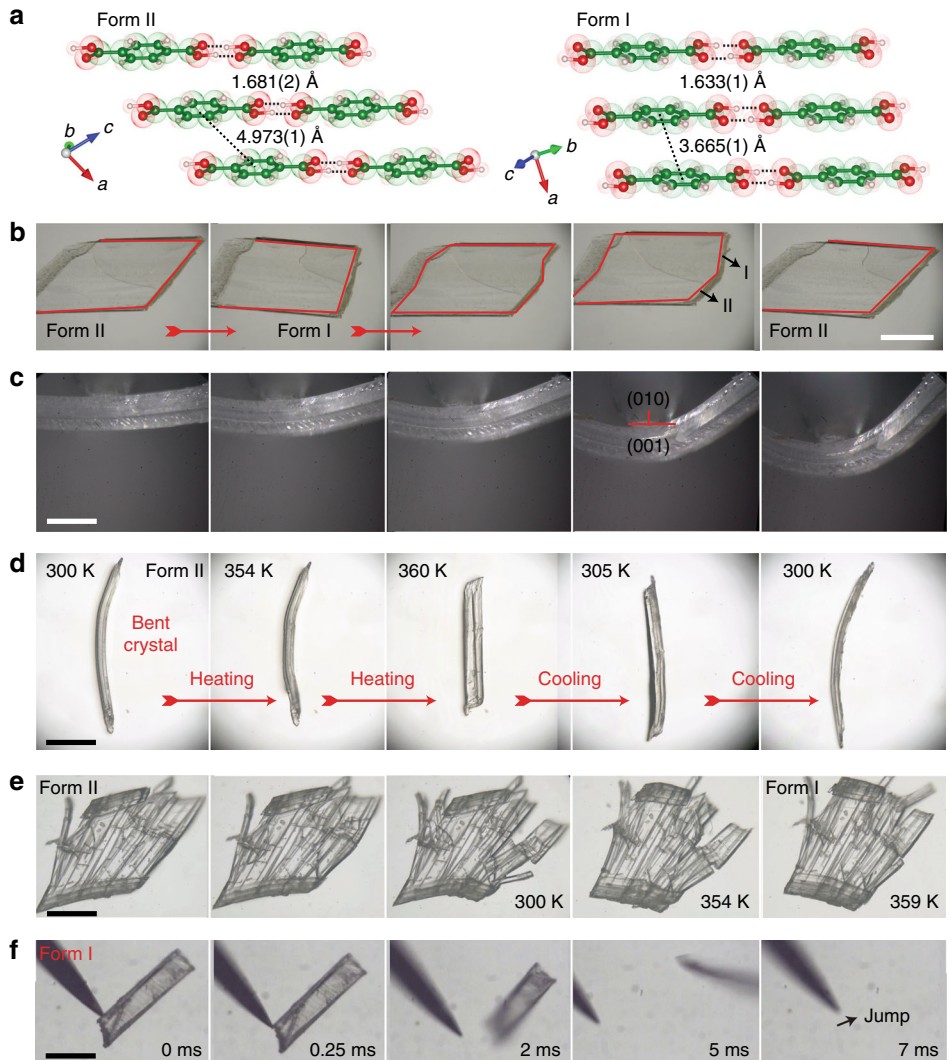

**Fig. 1** Bending, shape-memory and restorative abilities, and mechanosalient effect of TA crystals. **a** Polymorphism of TA. Layered structures of the crystals of the two polymorphs (forms I and II) are shown together with the intermolecular O···H distances and centroid-to-centroid distances between the aromatic rings from adjacent layers for either structure. The hydrogen bonds are shown as dotted lines. **b** Reversible thermally induced phase transition. A single crystal of form II is transformed to form I by heating, which is transformed back to form I by cooling. The edges of the colorless crystal are highlighted for clarity. Note the coexistence of domains of both phases in the intermediate stages. **c** Plastic bending. A crystal of TA is subjected to a pressure on its (010) face in a three-point geometry whereby it bends and retains the bent shape. Unlike other plastically bendable crystals, the striations develop at an angle to the longest axis of the crystal rather than along the axis. **d** Shape-memory effect. A straight crystal of form II is mechanically bent, and then taken over the phase transition by heating from 300 to 360 K, whereby it recovers its straight shape. The same crystal is then cooled to 300 K and it recovers its bent shape. This shape change can be repeated several times without deterioration. **e** Shape-restorative property. A crystal of form II was heavily damaged by applying pressure on its (010) ace. After heating to 359 K, the crystal appears to have partially restored its integrity due to phase transition and realignment of the fragments. Note the slightly different shape of the crystal after the phase transition. **f** Mechanosalient effect. An unrestrained crystal of form I obtained as a metastable form after heating and cooling a crystal of form II is contacted with a metal needle. The crystal springs off within 5 ms due to the strain that develops in its interior. Scale bars: **b** 800 μm; **c** 500 μm; **d** 600 μm; **e** 800 μm; **f** 500 μm

alternative heating and cooling, and no deterioration of the crystal was observed (Supplementary Fig. 9, Supplementary Movie 12). Mechanical bending of form II crystals can be performed at different locations simultaneously on both faces (010) and (0$\bar{1}$0) to deform the crystal into an S-shape, and the resulting crystal also exhibits shape-memory effect (Supplementary Fig. 10, Supplementary Movie 13). Altogether, these results demonstrate not only the reproducibility of the shape-memory effect, but also the mechanical robustness of molecular crystals and the potentials they may hold for organic shape-memory devices.

In addition to the shape-memory ability, crystals of this material that have been compressed or damaged on one of the two readily accessible faces, (010) and (001), are capable to partially restore their macroscopic integrity when they are heated above the phase transition (Fig. 1e). When a crystal of form II is lightly and uniformly pressed with a flat object on its (001) face, it develops cracks or it fragments parallel to its (010) face, but the resulting slabs remain conjoined (Fig. 2, Supplementary Movie 15). If the cracked crystal is taken over the phase transition by heating, each of the slabs is transformed to phase I, and they appear to slide back and partially restore a shape that resembles the original shape of the crystal. This shape-restoring ability was observed with multiple cracked crystals whose slabs or fragments tend to rearrange into a single entity (Supplementary Fig. 11, Supplementary Movies 16 and 17). Determination of the

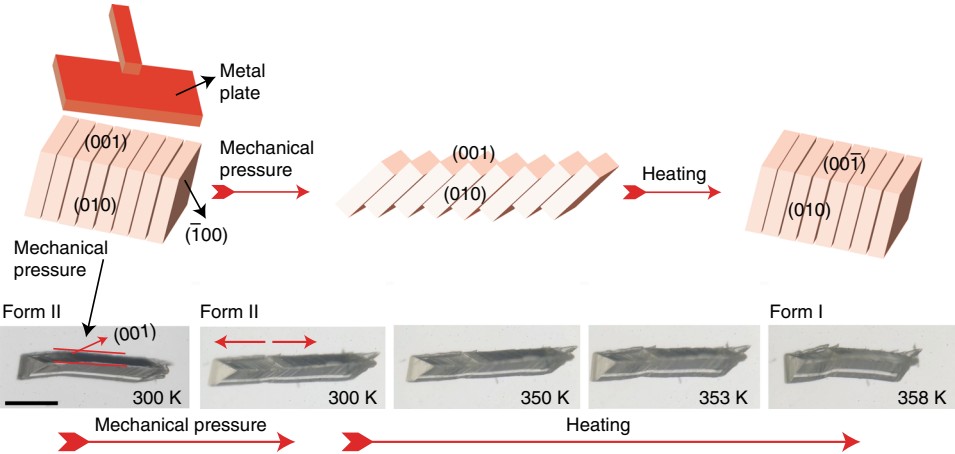

**Fig. 2** Mechanism of restoration of integrity of cracked crystal of TA by heating. A crystal of form II is damaged by applying pressure on its (001) face by using a metal plate whereby it develops multiple parallel cracks and is fragmented into slabs, but the fragments remain conjoined. Upon heating over the phase transition temperature, the fragments transition from form II to form I, and they tend to realign and restore the original shape of the crystal. Scale bar: 700 μm

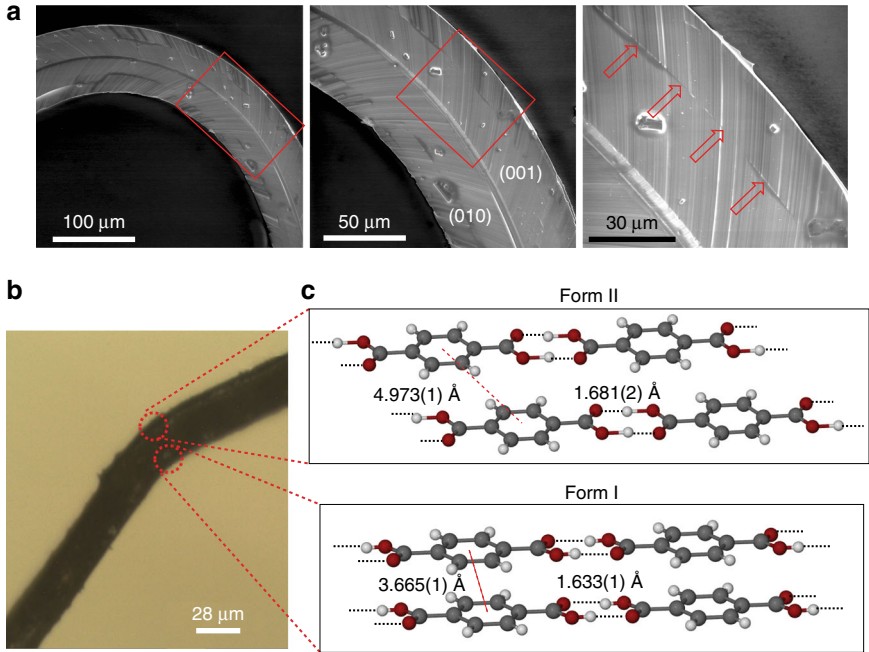

**Fig. 3** Electron microscopy imaging and structure analysis of a bent TA crystal. **a** Scanning electron microscopy (SEM) images of the crystal after bending with striations from the grain boundaries visible on the surface. The line that runs across the center of the crystal (marked with red arrows) corresponds to the habit plane (inter-phase boundary) between the two phases. **b** Optical image of a bent crystal of TA analyzed by microfocus X-ray diffraction. The approximate locations where the structure was determined are marked with circles. **c** Crystal structures determined in the circled regions of the bent crystal shown in panel (**b**). The lengths correspond to the centroid–centroid distances between the adjacent aromatic rings from two molecular strips and O⋯H distances between two hydrogen-bonded molecules

unit cell at different locations of the shape-restored crystals confirmed that they were completely transformed to form I (Supplementary Fig. 12, Supplementary Table 2). Moreover, even crystals that are heavily damaged with a pointy metal object and appear strongly disintegrated can be heated to partially recover their initial shape (Supplementary Fig. 13, Supplementary Movies 18 and 19). As established with XRD, the unit cell of such crystals was also invariably consistent with that of form I (Supplementary Figs. 13 and 14, Supplementary Table 3). In all cases of mechanically damaged and converted crystals the high-temperature phase was identical with form I, and the phase

identity was additionally confirmed with powder XRD (Supplementary Fig. 15).

**Microfocus XRD analysis of a bent crystal.** Figure 3a shows a close-up SEM image of the (001) face of a bent crystal of form II TA recorded normal to the plane of bending. Contrary to plastically bent crystals of other molecular solids, which typically develop striations and delaminate along the longest crystal axis[38], the surface of TA shows parallel striations that are at an angle of 130° to the longest axis. These striations correspond to grain

boundaries that evolve during bending and reflect the changing orientation of the crystal lattice across the deformed region. A single continuous phase boundary, highlighted with arrows in Fig. 3a was observed parallel to the crystal length, indicating that a habit plane was formed between two crystal domains that runs along the long axis of the crystal (additional SEM images are available from Supplementary Fig. 16).

The bent crystal of form II was analyzed by using microfocus XRD with synchrotron radiation. The concave and convex side of the bent region had discrete, well-defined diffraction patterns similar to that of the straight section of the crystal (Supplementary Figs. 17-19), and confirmed that during bending the crystal has retained its long-range structural order. However, while the unit cell parameters on the convex side were consistent to those of the original form II, the unit cell on the concave side was identical with that of form I. Complete diffraction data were collected at two points on the convex and the concave side at the kink of the bent crystal (Fig. 3b, Supplementary Table 4). Refinement of the crystal structure at the two points further confirmed that the two phases coexist in the bent crystal and their structures are identical to those of the pure phases (Fig. 3c, Supplementary Fig. 20). This result presents evidence that the inner side of the crystal has undergone phase transition during bending as a result of application of local pressure, and that the two phases co-exist in the bent crystal. Unlike the short time intervals of coexistence of the two phases during the thermosalient transition (seconds) and mechanosalient transition (milliseconds), the two phases in the bent crystal appeared to be indefinitely stable at constant (ambient) temperature. We hypothesize that the retention of crystal integrity may be attributed to the identical crystal symmetry of the two phases (triclinic, space group $P\bar{1}$) and the marginal difference in their unit cell volumes of only 0.07% although there is a significant difference in the individual unit cell axes (Supplementary Table 4). The results present evidence of a phase transition induced by bending in a crystal that has a metastable phase, and this is a qualitatively different mechanism for plastic bending compared to the more common delamination and sliding of layers along defects during plastic bending.

**Mechanism of plastic bending by partial phase transition**. The coexistence of the two phases in the bent region provides basis to establish an alternative mechanism for plastic bending of molecular crystals that can also explain the shape-memory and shape-restorative properties of this material described above (Fig. 4). Elongated crystals are usually bent by applying force at the two opposite termini of the crystal, or by the three-point method where the crystal is supported perpendicular to its length at both ends from one side and pressed at the middle section from the opposite side. The ensuing deformation occurs as a result of the applied shear strain which generates compressive and tensile stresses on the opposite faces of the crystal. This results in evolution of differential strain, ultimately generating a bending moment and the crystal bends.

Depending on the ability for shape recovery after the local pressure has been removed, application of pressure in either case can result in plastic or elastic bending (Fig. 4). The elastic bending is accompanied by expansion of the crystal and immediate restoration of the initial shape after the local pressure has been removed. In an ideal case, the angles between the crystal faces in an elastically bent beam are closely preserved (Fig. 4b). On the other hand, the *plastic* bending is regularly accompanied by sliding of molecular layers, whereby the individual layers glide but remain in contact with each other[35-39,53]. This is often observed as evolution of striations on the crystal surface perpendicular to the plane of bending (Fig. 4c; Supplementary Note 2). If such delaminated plastically bent crystal is bent and straightened several times, it usually folds and crinkles, and the slabs are partially physically separated due to the inability of the slabs on the convex side to fully stretch and realign. The delaminated layers can be visualized by rupturing the bent crystal[4]. Based on several instances of plastically bendable crystals having $\pi$–$\pi$ stacking interactions with molecular tapes or columns along the bending directions, the ability for plastic bending was ascribed to anisotropy in the crystal packing[35-39,53]. There is more recent evidence, however, that crystals with isotropic interactions can also be bent without fracture, indicating that structural anisotropy is not a sufficient prerequisite for plastic bending[10].

A common motif in the structures of the two forms of TA are infinite tapes of hydrogen-bonded acid molecules (O···H = 1.681 (2) Å in form II and O···H = 1.633(1) Å in form I; Fig. 3c). The tapes are linked into sheets by C–H···O interactions, and the sheets are stacked and interact with each other through $\pi$–$\pi$ interactions (Supplementary Fig. 20). The structures of the two forms differ only in the relative position of the adjacent layers, which are offset better in form II, with shortest inter-centroid distances between the phenyl rings of adjacent tapes of 4.973(1) Å and 3.665(1) Å in form II and I, respectively (Fig. 1a, Fig. 3c). The similarity of the two structures indicates that the layers can glide over each other, while they remain stacked. As established by nanoindentation, the crystals of both polymorphs are generally soft (Supplementary Figs. 7 and 21). The Young's modulus of form II is 6.2 ± 0.7 GPa and its hardness is 0.33 ± 0.05 GPa. Form I is comparatively softer in nature; its Young's modulus is 0.6 ± 0.2 GPa and its hardness is only 0.05 ± 0.03 GPa. The relatively softer nature of form I could compensate for the local pressure that must have developed after bending on the concave side of the crystal, and could facilitate the coexistence of form I and form II in the respective concave and convex regions of the bent crystal. The critical strain for bending, based on several crystal samples, was found to be 2.5 ± 0.2%; below this strain no phase transition occurs (Supplementary Fig. 22). The minimal force required to induce the phase transition of form II to form I by bending of the crystal obtained from the force–displacement profile along with the critical strain and averaged over eight crystals was determined to be 92 ± 5 mN (exemplary force–displacement and stress–strain curves are shown in Supplementary Fig. 7).

The mechanism of bending can be visualized better by analyzing the structure of the two forms in the bent crystal that can roughly be approximated by a bimorph (Fig. 5a, b). In form II, the carboxylic acid dimers form infinite tapes along the crystallographic direction [001]. The tapes are further stacked perpendicular to the (010) face and interact with each other through $\pi$–$\pi$ interactions along the crystallographic $a$ axis. As is schematically shown in Fig. 5b, crystals of form II TA are plastically bent perpendicular to their face (010) or (0$\bar{1}$0). This causes sliding and rotation of the molecular tapes in form II to nearly 30° on the concave side, where they reform the sheets along the crystallographic $c$ axis, and the concave part of the crystal is transformed to form I. The mismatch of the two unit cells, which—similar to their coexistence as alternating layers in a straight partially converted crystal—remain attached at the phase boundary, stabilizes the bimorph and the bent shape of the crystal is retained (Fig. 4d). When the bent crystal is heated over the phase transition, the form I on the concave side is transformed back to form II, and the unit cells in the two portions of the crystal become identical. This appears as partial or complete straightening of the crystal, and explains the shape-memory effect (Fig. 5c). The small remaining curvature can be attributed to residual permanent plastic deformation caused by defects introduced during the application of force to the crystal.

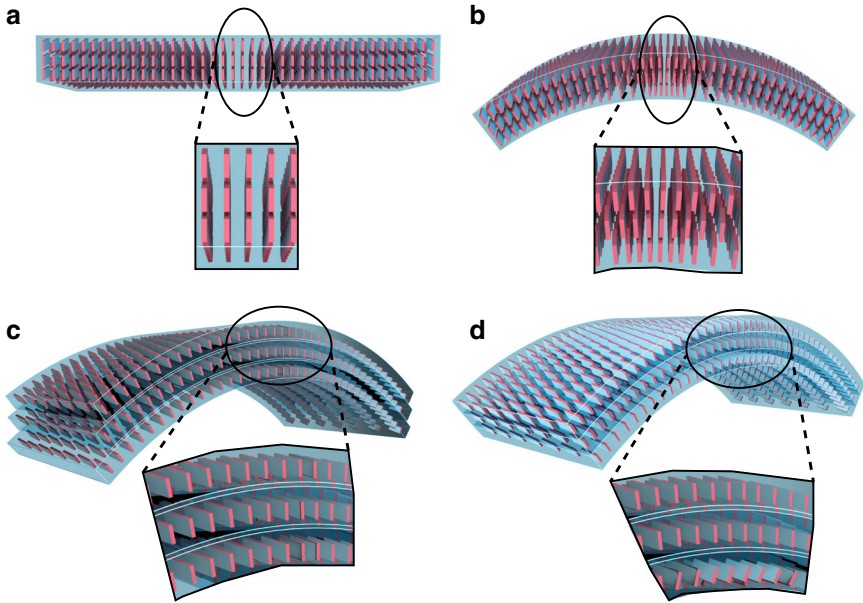

**Fig. 4** General mechanism for elastic and plastic bending of molecular crystals. **a** Sketch of a straight crystal before bending. The molecules are cartooned as a three-dimensional grid of blocks. **b** Mechanism of elastic bending. Elastic bending of the crystal occurs by significant expansion and separation of the molecules on the convex side, and relatively smaller compression on the concave side of the crystal. The angles between the crystal faces at the crystal termini are ideally preserved. **c** Mechanism of plastic bending by delamination. Plastic bending usually occurs by partial delamination along the crystal length whereby slabs of the crystal slide atop each other but remained bound to each other. In effect, the interfacial stress exerted during bending induces small expansion of the individual layers, although the deformation of each layer may still be detected. If the crystal is relatively short, the angles between the faces of the crystal at its termini may change; if the crystal is long, the delamination is not normally expected to proceed throughout the entire crystal. **d** Mechanism of plastic bending by phase transition. Application of localized stress induces a partial phase transition on the concave side of the crystal, and the two phases coexist in the bent region. One of the phases can be converted to the other by heating or cooling, and this results in apparent partial recovery of the straight shape of the crystal, similar to the shape-memory effect

## Discussion

Bending of simple, homogeneous objects such as beams is one of the most fundamental deformations from the mechanical engineering perspective, however, the mechanistic and structural details of this property have not been established in their entirety for bendable molecular crystals. Mechanical compliance of organic crystals is central to any future applications that would involve deformation during operation such as flexible organic electronics, organic optical waveguides, all-organic and soft robotics, and microfluidics. However, the increased concentration of defects and delamination during plastic bending, combined with crinkling upon straightening currently stand as impediment to utilization of these materials. The distinct mechanism of bending described here shows that unlike what has been established for other molecular crystals, plastic deformation of organic crystals does not have to necessarily occur by delamination. Indeed, the bimorph mechanism detailed here is endowed with greater mechanical compliance and reversibility compared to the common delamination mechanism for plastic bending. The apparent restoration of the shape of bent crystals and of the macroscopic integrity of cracked, unseparated crystals of TA are counter-intuitive and atypical properties for organic solid state that cannot be explained by the currently available models for elastic and plastic deformation of molecular crystals. Here, we provide evidence that these effects are related to a combination of pressure- and temperature-induced conversions between two phases, each of which can exist as a metastable phase beyond the phase transition temperature. Scanning electron microscopy and microfocus XRD using synchrotron radiation provided direct evidence that a crystal of one of the forms of TA can be partially converted to the other by applying pressure, and that both phases coexist at the kink of the mechanically bent crystal. The minimal

force required to induce the transition of the stable form to the metastable form by bending was found to be several tens of millinewtons. Heating or cooling over the phase transition results in complete conversion and accounts for the partial or complete recovery of the straight and bent shape of the crystal, akin to the shape memory effect that is known with some polymers and inorganic materials. The tendency of splintered or cracked crystals to apparently realign and join upon heating is a consequence of the phase transition and shape change of the individual fragments. The TA crystal provides example of pressure-induced phase transition in a bending crystal where the two phases coexist at ambient conditions and establishes a hitherto unreported mechanism of crystal bending. It is likely that this mechanism can account for shape-memory and restorative effects of other organic crystalline materials that are increasingly observed, but are almost never explained.

## Methods

**Materials**. Crystalline TA (CAS No. 100-21-0) was obtained from Sigma-Aldrich and used as received. Crystals of form II were grown under hydrothermal conditions as reported previously[4].

**Microscopy**. The thermal behavior of a bent crystal of TA was observed with a Q32634 hot-stage Q-imaging microscope (Linkam) equipped with a temperature-controlled stage THMS600-PS. Bent crystals of form II were heated to 318 K at a constant heating rate of 8 K min$^{-1}$. High-speed recordings were obtained with a HotShot 1280 CC camera (NAC) mounted on a Stereozoom SMZ745T trinocular stereoscope (Nikon). The video showing the transitions of form I to form II was recorded at a speed of 1500 frames per second. The SEM images were obtained using QUANTA FEG 450 electron microscope (Thermo Scientific) with primary electron energy of 10 kV.

**Microfocus XRD**. The synchrotron μ-XRD data were collected at beamline I19 of the Diamond Light Source using a dual air-bearing fixed–χ diffractometer equipped

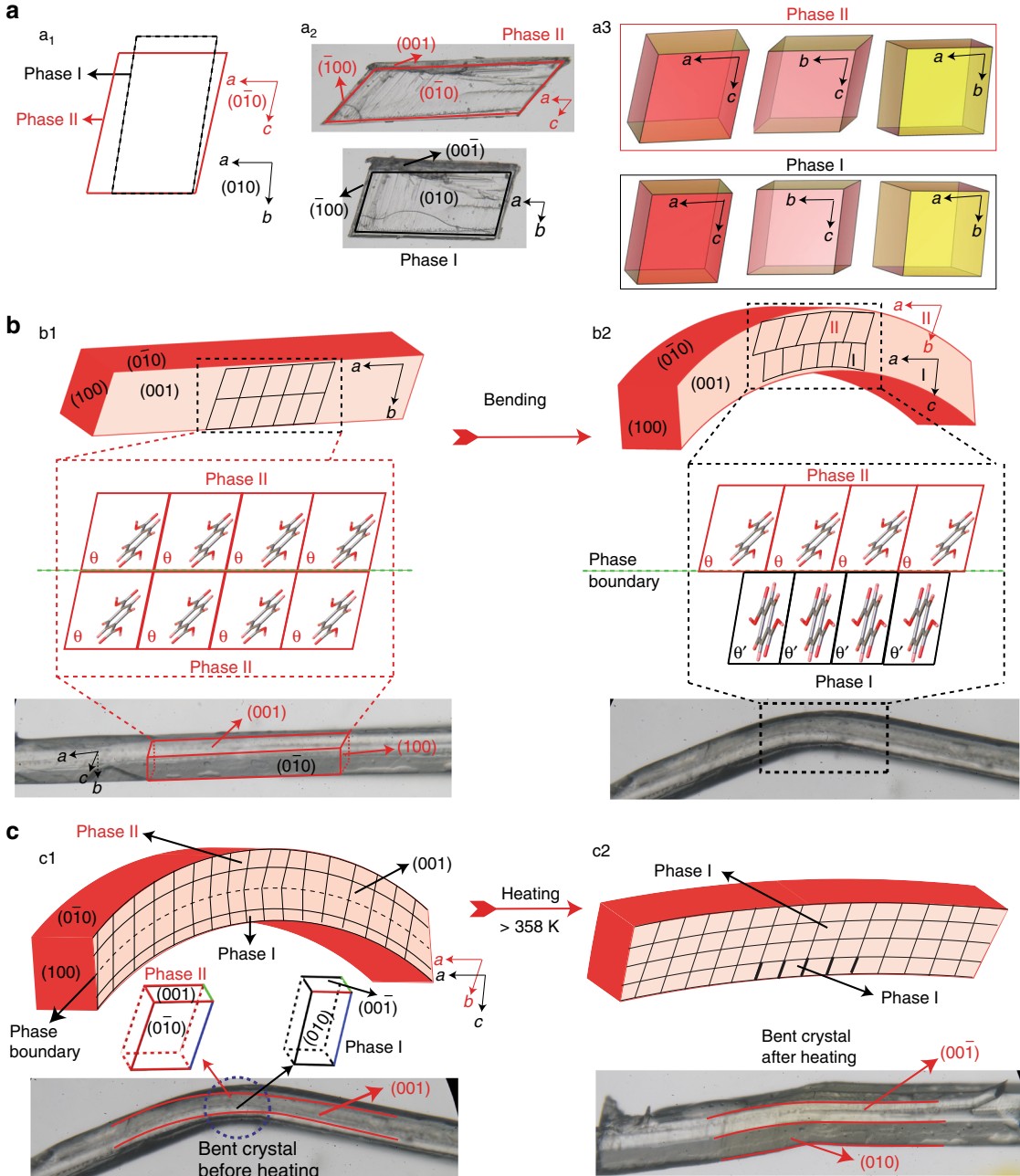

**Fig. 5** Mechanism of phase transition upon bending and shape-memory effect of a TA crystal. **a** Transformation of the crystal between forms II and I, and relationship between the unit cells in the two phases. (a1) Transformation between the unit cell orientations of form II and form I viewed normal to the crystallographic faces (0$\bar{1}$0) and (010). (a2) Images of the crystal in the two phases with face indices. (a3) Variation in the dimensions and shape of the unit cell, shown as different views. **b** Mechanism of transformation of the two-dimensional crystal lattice (cartoon) during the phase transition induced by application of local pressure including the structures of the two phases and the molecular orientation in each phase. (b1) Crystal of form II and schematic of its lattice before bending. The straight crystal of form II can be bent by applying pressure on its (010) or (0$\bar{1}$0) face. The lattice here is shown along the crystallographic [100] direction. (b2) Crystal and schematic of its lattice after bending. The bent crystal is a bimorph with two coexistent phases in the bent region separated by a phase boundary. **c** Mechanism of the shape-restorative effect. (c1) A bimorph in the kink of the bent crystal, shown with its lattice and unit cell. Unit cell orientations of both form II and form I in the bent region. (c2) Partial restoration of the straight shape of the crystal, and schematic representation of the underlying phase transition to form I. Further details on the structural relation between the two phases are available from Supplementary Fig. 27

with a PILATUS 2 M detector. A double-crystal Si(111) monochromator was used to select a wavelength of 0.6889 Å (17.997 keV). The data were collected with a detector distance of 160 mm with $\varphi$ and $2\theta$ were fixed at 90° and at 0°, respectively, while $\omega$ was in the range from 170° to 180°. The data collection and processing were carried out using CrysAlisPRO[54] software. Absorption correction was performed using spherical harmonics. The crystal structures were solved by direct methods and refined by full matrix least squares using SHELXL[55] and OLEX2 software[56].

**Nanoindentation**. The nanoindentation measurements were performed on form II and form I with an Agilent G200 nanoindenter equipped with an XP head, using Berkovich diamond indenter. Indentation was performed using the continuous stiffness method to a depth of 1000 nm with a strain rate of 0.05, an amplitude of 2 nm, and a frequency of 45 Hz[57]. The stiffness and the geometry of the tip were determined by using Corning 7980 silica reference sample (Nanomechanics S1495-25) before performing indentation on form II and form I. In order to ensure that the tip was fully engaged, the modulus was measured between 200 and 1000 nm.

**Mechanical testing**. A tensile tester, SEMTester DAQ-linear (model 8000-0014, MTI Instruments), was used for three-point bend tests at room temperature. The tester was equipped with a 5 N load cell and a three-point bending apparatus with a 1.5 mm span. A crosshead speed of 0.05 mm min$^{-1}$ was applied.

## Data availability

The X-ray crystallographic coordinates and other details for structures reported in this study have been deposited at the Cambridge Crystallographic Data Centre (CCDC), under deposition numbers 1854403 and 1854404. These can be obtained free of charge from The Cambridge Crystallographic Data Centre via www.ccdc.cam.ac.uk/data_request/cif.

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

## Acknowledgements

We thank New York University Abu Dhabi for the financial support for this work. This research was partially carried out using the Core Technology Platform resources at New York University Abu Dhabi. We sincerely thank Dr. Liang Li, Dr. James Weston, and Dr. Patrick Commins for technical assistance. We gratefully acknowledge Diamond Light Source (DLS) Oxford, United Kingdom, for the beamtime to perform the synchrotron experiments at I19 beamline (project ID: MT15848-1). We also thank Dr. David Allan and Dr. Sarah Barnett for their technical support at DLS.

## Author contributions

P.N. conceived the study. D.P.K. prepared the crystals and E.A. performed the mechanical tests. E.A and M.W. collected the synchrotron data. D.P.K. and E.A. solved and analyzed the synchrotron XRD data and performed phase transition experiments. D.P.K. performed nanoindentation. D.P.K., E.A. and P.N. analyzed the phase transition in the bent region and established the mechanism. The paper was written with the contributions from E.A., D.P.K. and P.N. All authors have given approval to the final version of the paper.

## Additional information

**Competing interests:** The authors declare no competing interests.

