## [Peer Review File · Nature Communications]

Reviewers' comments:

Reviewer #1 (Remarks to the Author):

Key results:

The authors have investigated some of the mechanical properties of crystals of terephthalic acid. They find that upon application of strain within the crystal they can induce a phase change which leads to deformation of the crystal. The authors claim it is reversible and a new mechanism of bending.

Validity: Does the manuscript have flaws which should prohibit its publication? If so, please provide details.

The claims of the authors are simply not supported. There are many definitional issues throughout the manuscript and statements that are factually incorrect. These statements are not supported by references or experiments. The biggest flaw and most significant is a requirement that the authors demonstrate that these are single crystals - indeed their microfocus diffraction experiments show that they are not single and probably not crystals. The propose a mechanism but it is mere conjecture it has not been established by experiment. Insufficient experimental details have been given to allow the reproducibility of the results. The

Originality and significance: If the conclusions are not original, please provide relevant references. On a more subjective note, do you feel that the results presented are of immediate interest to many people in your own discipline, and/or to people from several disciplines?

This is not new or original. Miyamoto et al. published a very similar compound showing very similar properties in 2014 (Angew. Chemie., 2014, 53, 6970). The study is far superior to the present one.

Data & methodology: Please comment on the validity of the approach, quality of the data and quality of presentation. Please note that we expect our reviewers to review all data, including any extended data and supplementary information. Is the reporting of data and methodology sufficiently detailed and transparent to enable reproducing the results?

see above - for example there are no details of the Xray experiments, other than that they were collected at Diamond.

Conclusions: Do you find that the conclusions and data interpretation are robust, valid and reliable?

No.

References: Does this manuscript reference previous literature appropriately? If not, what references should be included or excluded?

No, there is over-referencing of the authors' own work.

Clarity and context: Is the abstract clear, accessible? Are abstract, introduction and conclusions appropriate?

No - they are completely misleading. The paper appears to be selling results that are not substantiated - the phrase mutton dressed as lamb comes to mind.

Reviewer #2 (Remarks to the Author):

The authors presented interesting results on the shape-memory effects observed during mechanically and thermally induced polymorphic transformations between two forms of terephthalic acid. I am really excited by the scope of the work but I think it is somewhat narrow in scope for folks outside of this area and authors should improve the introduction/conclusions to make it more appealing to others outside of the field. How general is this result? Is it transformative and if yes, give some concrete examples. I also would like to see discussion on the change in mechanical properties between the two polymorphs and how it may contribute to the apparent partial shape-restoration effect. Below please find several additional specific questions and suggestions:

1. Figure 1D – the authors claim the partial restoration of the original crystal shape, I would like to see more results and analysis to support this conclusion. In particular, please add results using additional crystals and include analysis on the extent of recovery, does it depend on the extent of the initial mechanical bending? Can similar effect be observed without heating, i.e. can form II after bending transform to Form I at 300 K, and if yes, does that lead to change in the shape? What happens if other crystal shapes are studied, i.e. more prism-like or rods with different aspect ratios.
2. What would happen if authors try to do these experiments multiple times by going back and forth between two polymorphs? For example, Fig1D, last panel to the right, can form I crystal be converted to form II again? And if yes, can mechanical force now be applied to opposite direction so that mechanical bending is opposite to how it was done initially?
3. Authors mention Young's modulus of form II, would be valuable to include result for form I as well. I also recommend adding hardness results here as well, in particular since significant plastic deformations take place as a result of applying mechanical force. Can change in the Young's modulus between two forms be part of the explanation for the observed shape change, for example difference in the mechanical properties of two forms changes stress/strain which is build up during the transformation, that could explain observed partial recovery. I think further experiments + corresponding discussion would strengthen the paper in addition to primarily structural arguments.
4. Figure 5, almost no discussion in text/caption, I had difficult time following it, I suggest expanding text + corresponding figure caption or try to simplify it

Alexei V. Tivanski, Ph.D.
Associate Professor of Chemistry
Department of Chemistry
Room E272 CB
University of Iowa
Iowa City, IA 52245
Phone: 319-384-3692
Group homepage: <https://chem.uiowa.edu/tivanski-research-group>

Reviewer #3 (Remarks to the Author):

Mechanically responsive molecular crystals have been a topic of significant interest in the last few years, with all sorts of surprising properties being observed (twisting, bending, elongation, jumping, etc). This really nice paper demonstrates that applying mechanical force to terephthalic acid crystals results in bending and partial phase transition to a different polymorph. However, heating reverses the phase transition and at least partially restores the original crystal shape, which is reminiscent of shape-memory polymers. While one can find other examples of reversible bending and mechanical responses induced by phase changes in the literature, I believe the combination of mechanically induced plastic bending that retains its shape which can then be restored thermally is pretty unique.

The second key aspect that stands out in this work is that they were able to provide strong evidence for the proposed mechanism by obtaining crystal structures on the concave and convex sides of the bent crystal to demonstrate the partial phase transformation, and they argue convincingly how these changes in crystal packing relate to the rod deformation.

The combination of the unique crystalline behaviors and the strong structural evidence makes this paper worthy of publication in Nature Comm. I have only a couple very minor comments:

- Is phase II of TA generally more stable under isotropic/hydrostatic pressure, or does the phase transition only result from the anisotropic nature of the local forces being applied here?
- The paper notes large differences (1.80 vs 1.44 Ang) in the hydrogen bond lengths for the two polymorphs, but the crystal structure details for the COOH groups in the CIF files are suspect. In both polymorphs, the intermolecular O..O distances are very similar at 2.62 and 2.63 Angstroms. However, in one case, the O-H covalent bond is surprisingly short (0.82 Ang), and in the other it seems too long at 1.19 Ang. This then translates to the large differences seen in the H..O hydrogen bond lengths. I am not a crystallographer, and I recognize that H atom positions can be difficult to infer, but this still seems wrong (or at least something to discuss). None of this will likely change any of the important arguments, but they should look into the issue.

Reviewer #4 (Remarks to the Author):

The authors report synchrotron microfocus X-ray diffraction analysis of the bent crystal of terephthalic acid (TA) that has two coexisting polymorphs and suggest a new mechanism for bent crystals as a mechanically induced phase transition. The introduced mechanism provides some insight on shape memory and restorative effects seen in other molecular crystals. However, many of the results discussed in the paper including the shape memory and self-healing effects of this crystal have already been discussed in their previous work (J. Am. Chem. Soc. 2016, 138, 13298). Specifically, most of the results presented in Figures 1–3 of this work have been discussed before in detail. There are numerous additional results that should be added overall and some of the arguments are not well supported. Thus this work requires major revision by addressing all questions and points suggested below before being considered for publication in Nature Communications.

- Abstract and conclusion generalize the mechanism seen in TA crystal to soft crystals and organic solids, but this study only focuses on the TA crystal (which most of the results have been discussed in JACS 2016). There is not enough evidence to support that what happens in the TA crystals is generalizable to other molecular crystals.
- Phase identification in the bent region requires much more detail. This is the critical part of the paper. The data presented does not fully support the conclusion that the bent region is bimorphic. The microdiffraction results presented in Figure S5 is not of very high quality (showing doublets). Also the result is only presented in the compressed region, a single diffraction experiment, but not over the entire bent region as a mapping result. Mapping is needed to show how the polymorphs vary: is it continuous or an abrupt change from form I to II? The diffraction image presented needs to be indexed and the unit cell refinement results need to be presented with R values (Table S1 does not contain this info).

Section "Phase transition and shape restorative effects"

- Timescale on transition: In the first paragraph, when they indicate that the crystals "do not transition sharply", "rapid reshaping upon mechanical stimulation", etc. what does that mean? Is it 1st order or 2nd order transition? Need to show DSC results which show 1st order transition
- o On a related note, Supplementary Videos do not have time stamps or scale bars
- "Twinned crystals" can indicate that the two forms are twins, indicating that they are of same polymorph. In this work, they are form I and II, and are different polymorphs
- From Supplementary Movie 1, it is hard to believe that the crystal springs off due to phase transition rather than force of the sharp object pushing down on the edge of the crystal. In the video the partial transition from I to II already takes place in the beginning before jumping off. Is

there proof that the crystal completely transitioned to form II? Also, this phenomenon was already discussed in JACS 2016, it is unclear what the purpose of mentioning it again here is.

- Figure 1B: Is there structural proof that the region that the sharp object touches is Form II? Was micro X-ray also done in that region? If so, it should be discussed, if not, there should be supporting evidence that the transitioned region is indeed the stable Form II, using other methods to cross validate such as spectroscopy.
- Figure 1D: This is identical to Figure 4A of JACS 2016, where they bend the crystal and heat. It is unclear what new information this figure adds to this work. Also, what happens to the crystal after cooling? Would the crystal transform back to Form II? Would you be able to start from Form I and induce mechanical bending as well or does bending only occur on Form II? Need more details.
- Figure 1E: Supplementary Video shows clear restoration, but from the figure it is hard to see. Also, need to show evidence that after heating, the crystal is fully transformed to Form I. Was X-ray repeated on the crystal after heating? Since it did not completely restore is it possible that it may not be completely Form I? Heating alone can't guarantee that the crystal has completely transformed to Form I.
- Figure 2: Again, how is it proven that it is entirely Form I at 358K? The bottom figures and the supplementary movie seems to show that maybe the crystal has partially transformed from Form II to I, as the phase propagation does not seem to go through all the way. Does it eventually transform completely? Is phase transformation affected by defect density or location of defects caused by the metal plate?

Section "Microfocus X-ray diffraction analysis of a bent crystal"

- As indicated in manuscript, Supplementary Figure 5 shows long range order, but also shows several diffraction peaks close together, which can indicate twinning or gliding of the different layers. Explanation of what they are is needed.
- The manuscript describes Supplementary Table 1 as the result of refinement on the bent convex and concave side crystals, but the title of the table is: Crystallographic data and refinement details of terephthalic acid structures refined from the same crystal at different temperatures. It seems like the title is labeled incorrectly.
- It would be interesting to see if there is a critical strain for forming I on the compressed concave side of the crystal. With less than critical strain, then both sides of the crystal even after bending may still be Form II.
- It would be interesting to see if the crystal integrity can still be maintained if the crystal is bent the opposite way.
- A comparison of the bent crystal Form II to the pristine Form II at same temperatures would be interesting. Since the convex part still experiences expansion as Form II, does it have expanded/contracted unit cell values when compared to the pristine case?

Section "Mechanism of plastic bending by partial phase transition"

- Authors described in last paragraph before Figure 5 "This causes sliding and rotation of nearly 30° of the tapes on the concave side, ..." which angle is this referring to? The author never described in the manuscript.
- Also authors mentioned in the same paragraph that: "The slight mismatch of the two unit cells, which—similar to their coexistence as alternating layers in a straight partially converted crystal—remain attached at the phase boundary, stabilizes the biphase and the bent shape of the crystal is retained (Figure 4D)."

As authors show the lattice structure in Figure 5A the mismatch at the phase boundary looks significant (for instance around 26.3 % decrease in a-axis length); which do not mean slight mismatch of the two unit cells. And in Figure 5B on the right schematic exaggerated the lattice dimension of Phase I to seem like it matches to that of Phase II.

- Incorrect statement right before Figure 5: "When the bent crystal is heated over the phase transition, the form I on the concave side is transformed back to form II" (it should be form II to form I)

Response to the comments from the reviewers

We thank all reviewers for the valuable comments which have contributed significantly to improve the quality of the manuscript. We considered all comments, and we tried to address them to the best of our ability. Unless stated otherwise, all numbers of the figures and supplementary materials refer to the *revised* version of the manuscript. Together with this submission we have provided marked copies of the main text and the Supplementary Materials where the changes to the original version have been marked. For convenience, in what follows, the original comments from the reviewers are highlighted in *blue color*, our response is provided in black color, and the text that was modified or added to the manuscript is marked with *red color*.

Response to the comments from Reviewer #1

Comment: *Key results: The authors have investigated some of the mechanical properties of crystals of terephthalic acid. They find that upon application of strain within the crystal they can induce a phase change which leads to deformation of the crystal. The authors claim it is reversible and a new mechanism of bending.*

The claims of the authors are simply not supported. There are many definitional issues throughout the manuscript and statements that are factually incorrect. These statements are not supported by references or experiments. The biggest flaw and most significant is a requirement that the authors demonstrate that these are single crystals - indeed their microfocus diffraction experiments show that they are not single and probably not crystals. The propose a mechanism but it is mere conjecture it has not been established by experiment. Insufficient experimental details have been given to allow the reproducibility of the results.

Response to the comment: We appreciate the reviewer's assessment of our results, and in the response below we address their concerns.

We are quite perplexed with the reviewer's conclusion that the "microfocus X-ray diffraction experiments show that their crystals are not single crystals", particularly in view of the fact that three out of the four authors on this manuscript are trained crystallographers. The corresponding author of this manuscript has over 20 years of experience in small-molecule crystallography and has determined and published hundreds of crystal structures. One of the other crystallographers on this manuscript works in a crystallography beamline at a synchrotron, and the third is specialist for crystallography with in-house diffractometers.

We are quite confident, based on the strong and clear diffraction data with well-defined Bragg reflections, that our samples are single crystals. In the manuscript, we have already provided complete crystal structures determined and refined from the outer part of the crystal and the inner part of the crystal. This data were collected by using synchrotron X-rays. The details on the refinement are available from the Supplementary Table 4. The

same data have been deposited within the Cambridge Structure Database, and are accessible for inspection if needed. In the original version of the manuscript, we have also provided images of diffractions that confirm that the samples are single crystals. There is no better way to confirm the single crystal nature of these samples than the inspection of the primary diffraction images and fully refined crystal structures that clearly evidence that the material is in form of single crystals. In view of the collective experimental data, we do not agree that that the mechanism which we propose is unfounded, because we have provided firm experimental data that clearly support that mechanism.

We would also like to note that the definitions of the terms “single crystal” and “crystallinity” are out of scope of this work, because these terms are already well defined in the crystallography, they are commonly accepted, and can not be a subject of personal interpretation. We would like to draw the editor’s and the reviewer’s attention to a recent peer-reviewed publication where we have discussed this subject in greater detail, and based on a large amount of results on bent crystals from the literature: P. Commins, D. P. Karothu, P. Naumov, “Is a bent crystal still a single crystal?”, *Angew. Chem., Int. Ed.* 2019, in press (<https://doi.org/10.1002/anie.201814387>). The conclusion in this peer-reviewed publication is very clear: bent crystals are single crystals as long as they are not completely delaminated in separate crystals. We would like to reiterate here that the definition of a single crystal given by the International Union of Crystallography is very clear and can not be subject to different interpretations. The above article explains how it applies on bent molecular crystals, and we strongly recommend that the reviewer becomes acquainted with the definition and its interpretation. We do not find it necessary to redefine these established definitions in the current article, because they are available from the above article and the general crystallographic literature.

In response to the reviewer’s comment, we provide a copy of the conclusions paragraph of the above paper:

“Going to the question posed in the title of this article, among the articles that describe bending crystals, the amount of reports that provide thorough X-ray diffraction analysis of bending crystals that were single crystals before being bent is still surprisingly small and thus a general consensus on the single crystallinity of these compounds has not been established. The available data, however, clearly show—within the IUCr definition of a single crystal—crystals that bend elastically and crystals that bend plastically without physical separation are clearly single crystals after they have been deformed, the latter having expectedly higher concentration of defects. Single crystals where visible separation occurs during bending would normally be considered multiple crystals after the bending. Even when the layers do not separate after the first bending, plastic crystals are expected to partially delaminate during repeated bending and unbending; they may maintains single crystallinity after one cycle and gradually become more polycrystalline after successive bending cycles. We hope that this collection of results will clarify some of the confusion, both semantic and factual, with how mechanically complaint crystals should be regarded in future.”

The crystal nature of the terephthalic acid has been well established in the earlier literature which is cited in our manuscript. For example, see the article by Roger Davey: Davey, R. J. et al. Morphology and polymorphism in molecular crystals: terephthalic acid. J. Chem. Soc., Faraday Trans. 90, 1003–1009 (1994). Our samples of terephthalic acid conform to the earlier literature, and they are clearly single crystals, as can be also inspected from the diffraction images recorded from both forms provide below:

Terephthalic acid - Form II

Terephthalic acid - Form I

The details of the refined crystal structures can also be inspected from the crystallographic details in Supplementary Table 4, copied below:

Supplementary Table 4. Crystallographic data and refinement details of the two domains in a bent crystal of terephthalic acid refined from the convex (form II) and concave (form I) regions of a bent crystal

Bent region	Convex	Concave
Temperature / K	290	290
Polymorph	Form II	Form I
Formula weight	166.13	166.13
Crystal system	Triclinic	Triclinic
Space group	$P\bar{1}$	$P\bar{1}$
$a / \text{\AA}$	4.97280(10)	3.66490(10)
$b / \text{\AA}$	5.30140(10)	6.3683(2)
$c / \text{\AA}$	6.9746(2)	7.4304(2)

$\alpha / ^\circ$	72.653(2)	83.723(2)
$\beta / ^\circ$	74.880(2)	79.758(3)
$\gamma / ^\circ$	86.245(2)	87.345(3)
Volume / \AA^3	169.416(7)	169.569(9)
Z	2	2
Density / (g cm^{-3})	1.628	1.627
μ / mm^{-1}	0.125	0.125
F_{000}	86	86
h_{\min}, h_{\max}	-5, 4	-4, 4
k_{\min}, k_{\max}	-5, 5	-7, 7
l_{\min}, l_{\max}	-6, 6	-8, 8
No. of measured reflections	648	1423
No. of unique reflections	386	460
No. of reflections used	348	453
$R_{\text{all}}, R_{\text{obs}}$	0.09, 0.08	0.100, 0.099
$wR_{2,\text{all}}, wR_{2,\text{obs}}$	0.238, 0.234	0.343, 0.342
$\Delta\rho_{\min,\max} / (\text{e \AA}^{-3})$	-0.506, 0.450	-0.611, 0.812
Goof	1.179	1.290
CCDC No.	1854403	1854404

As we have explained in the manuscript, crystals of form II undergo a reversible phase transition. In the revised version of the manuscript we have included additional diffraction images that were collected at the concave side, the convex side and the straight section of the crystal. They are included as Supplementary Figures 17, 18 and 19. These images further confirm the single crystal nature of the bent crystal. For convenience, they are copied below (magnified images are available from the Supplementary Material).

Comment: *This is not new or original. Miyamoto et al. published a very similar compound showing very similar properties in 2014 (Angew. Chemie., 2014, 53, 6970). The study is far superior to the present one.*

Response to the comment:

The authors of this work are very well aware of the work of Miyamoto and Takamizawa. However, both the material and the phenomenon described in the above article are very different from those studied in this work (we would also like to refrain from commenting on one article as being “superior” to another one). Namely, the compound studied by Takamizawa and Miyamoto is terephthalamide, and is a so-called “superelastic” organic crystal. By applying shear stress on this crystal, the crystal transforms to another form. Once the stress is released, the crystal reverts to its original form. This behavior is completely different from the one that we describe in the current work.

The terephthalic acid (TA) crystals that we describe here show thermosalient and mechanosalient behaviors which are not observed with the above compound. Moreover, the form II crystals of TA show shape-memory and self-healing like properties. Most importantly, the crystals of TA can be bent plastically and they do not revert to their initial shape spontaneously. These bent crystals can be further reshaped into straight and bent shapes by applying force for several cycles without any deterioration. What we report in this manuscript is the first ever example where upon bending of form II TA crystal, the stable form II in the bent region is transformed to the metastable form I. This has not been reported before. Due to the large differences in the Young’s modulus and hardness of the two forms and the relatively soft nature of form I, the two forms can coexist as a bimorph in the bent region. We have characterized this bimorph in the bent region by using X-ray diffraction and scanning electron microscopy, in addition to other methods.

The figure below describes visually the difference between the two systems.

Comment: *see above - for example there are no details of the Xray experiments, other than that they were collected at Diamond.*

Response to the comment: We would like to bring to the reviewer's attention that all experimental details have been clearly described in great detail in the Methods section. A copy of this section is provided below for reference.

“Microfocus X-ray diffraction. The synchrotron μ -XRD data were collected at beamline I19 of the Diamond Light Source using a dual air-bearing fixed- χ diffractometer equipped with a PILATUS 2M detector. A double-crystal Si(111) monochromator was used to select a wavelength of 0.6889 Å (17.997 keV). The data were collected with a detector distance of 160 mm with ϕ and 2θ were fixed at 90° and at 0° respectively while ω was in the range from 170° to 180° . The data collection and processing were carried out using CrysAlisPRO⁴⁸ software. Absorption correction was performed using spherical harmonics. The crystal structures were solved by direct methods and refined by full matrix least squares using SHELXL⁴⁹ and OLEX2 software.⁵⁰

Comment: *Conclusions: Do you find that the conclusions and data interpretation are robust, valid and reliable?*

No.

Response to the comment: We would have appreciated if the reviewer elaborated in greater detail on why our conclusions and interpretation are not robust, however we regret that such details are not provided.

Comment: *References: Does this manuscript reference previous literature appropriately? If not, what references should be included or excluded?*

No, there is over-referencing of the authors' own work.

Response to the comment: We do not agree with the reviewer on this statement. Out of 51 references, only 14 are from the authors of this article, which accounts for 27% of the overall reference count. We believe that this portion of self-citation is acceptable, especially considering the previous activity of the authors and their contribution to this research field. We included both older and newer articles in the reference list. All other articles and other resources that are relevant to this work have been included in the reference list, considering the limitations with the number of references as per the journal's guidelines.

Comment: *Clarity and context: Is the abstract clear, accessible? Are abstract, introduction and conclusions appropriate?*

No - they are completely misleading. The paper appears to be selling results that are not substantiated - the phrase mutton dressed as lamb comes to mind.

Response to the comment: We are not clear of the exact meaning of this comment and comparison of our results to mutton and a lamb, but we do believe that the contents of our abstract clearly desalinates the contents and the main conclusions reached in this work. We would have appreciated, however, to have a constructive comment on why in the reviewer's opinion our results are not substantiated. In response to the reviewer's comments we have modified the abstract and the introduction, and we hope that the revised version reflects better the contents and the main conclusion from this study.

Response to the comments from Reviewer #2

Comment: *The authors presented interesting results on the shape-memory effects observed during mechanically and thermally induced polymorphic transformations between two forms of terephthalic acid. I am really excited by the scope of the work but I think it is somewhat narrow in scope for folks outside of this area and authors should improve the introduction/conclusions to make it more appealing to others outside of the field.*

Response to the comment: We thank the reviewer for the generally positive assessment of the contents of our manuscript, and for the constructive suggestions. All suggestions were considered and taken into account in the revised version.

We have revised the abstract, the introduction, and to a greater extent, the conclusions section in order to place the results within a more general context. We attempted to add to the generality of the results in the abstract of the original submission, where we state that “Heating over the phase transition temperature partially recovers their original straight shape and appears as a shape memory effect akin of that observed with polymers or metals.” in hope that the general audience will recognize the more general implications of the results presented in this work. Although we are very limited with word count, in the revised version we added text to highlight the possible applications of these materials. Below, we have copied the new version of the abstract:

“Molecular crystals can be bent elastically by expansion, or plastically by delamination into slabs that glide along slip-planes. Here we report that organic crystals can also be bent by a distinctly different mechanism, a mechanically induced phase transition that occurs without delamination and preserves the overall crystal integrity. While generally the shape of elastically bent crystals is restored immediately after removal of the external force and plastically bent crystals remain deformed indefinitely, crystals where two phases coexist effectively act as biphases and their phase uniformity can be recovered thermally—heating over the phase transition temperature partially recovers their original straight shape and upon cooling they revert to their bent habit—resulting in shape memory effects akin of those observed with some metal alloys and polymers. The phase transition also accounts for partial recovery of the integrity of mechanically damaged crystals. We anticipate that similar memory and restorative effects are common for other molecular crystals which have polymorphs that can coexist in the same crystal with low interfacial energy. The results demonstrate the advantage of using intermolecular interactions to accomplish mechanically adaptive properties with organic solids that bridge the gap in the materials space between mesophasic and inorganic materials.”

Comment: *How general is this result? Is it transformative and if yes, give some concrete examples.*

Response to the comment: We share reviewer’s sentiment that at first reading, the text might appear rather technical and narrow in scope. We would like to note that the main point of originality in this work is that this is the first example of a molecular crystal which shows phase transformation during bending. At ambient temperature, the resulting bent crystal remains a stable biphase and the bent region having form I and form II in the concave and convex regions, respectively. So far, we have not observed this phase transformation in the bent region in any other organic molecule, and we are not aware of another reported example. However, as we have also delineated in the revised text, we do expect that other crystals could undergo similar phase transition during bending:

Abstract: *“We anticipate that similar memory and restorative effects are common for other molecular crystals which have polymorphs that can coexist in the same crystal with low interfacial energy. The results demonstrate the advantage of using intermolecular interactions to accomplish mechanically adaptive properties with organic solids that bridge the gap in the materials space between mesophasic and inorganic materials.”*

Conclusions: *“The TA crystal provides the first example of pressure-induced phase transition in a bending crystal where the two phases coexist at ambient conditions and establishes a hitherto unreported mechanism of crystal bending. It is likely that this mechanism can account for shape-memory and restorative effects of other organic crystalline materials that are increasingly observed, but are almost never explained.”*

We are rather limited with the word count in the abstract and the introduction to provide more information of the general significance and broader impact of the results. Instead, we elaborated more on this in the conclusions section in the revised version.

“Bending of simple, homogeneous objects such as beams is one of the most fundamental deformations from the mechanical engineering perspective, however the mechanistic and structural details of this property have not been established in their entirety for bendable molecular crystals. Mechanical compliance of organic crystals is central to any future applications that would involve deformation during operation such as flexible organic electronics, organic optical waveguides, all-organic and soft robotics, and microfluidics. However, the increased concentration of defects and delamination during plastic bending, combined with crinkling upon straightening currently stand as impediment to utilization of these materials. The distinct mechanism of bending described here shows that unlike what has been established for other crystals, plastic deformation of organic crystals does not have to necessarily occur by delamination. Indeed, the bimorph mechanism established here is endowed with greater mechanical compliance and reversibility compared to the common delamination mechanism for plastic bending.”

Comment: *I also would like to see discussion on the change in mechanical properties between the two polymorphs and how it may contribute to the apparent partial shape-restoration effect. Below please find several additional specific questions and suggestions:*

Response to the comment: We thank the reviewer for this constructive suggestion. Although it was challenging exercise for one of the polymorphs, in the meantime we succeeded to measure the Young's modulus and hardness for both forms II and I by using nanoindentation. In the revised version of the manuscript we have included the discussion on the mechanical properties of both forms and their contribution towards the partial shape-restoration effect. The following text was added in the main text and Supplementary Figures 21 and 22 were added in the Supplementary Information:

“As established by nanoindentation, the crystals of both polymorphs are generally soft (Supplementary Figures 7 and 21). The Young’s modulus of form II is 6.2 ± 0.7 GPa and its hardness is 0.33 ± 0.05 GPa. Form I is comparatively softer in nature; its Young’s modulus is 0.6 ± 0.2 GPa and its hardness is only 0.05 ± 0.03 GPa. The relatively softer nature of form I could compensate for the local pressure that must developed after bending on the concave side of the crystal, and could facilitate the coexistence of form I and form II in the respective concave and convex regions of the bent crystal. The critical strain for bending, based on several crystal samples, was found to be $2.5 \pm 0.2\%$; below this strain no phase transition occurs (Supplementary Figure 22). The minimal force required to induce the phase transition of form II to form I by bending of the crystal obtained from the force-displacement profile along with the critical strain and averaged over eight crystals was determined to be 92 ± 5 mN (an exemplary force-displacement and stress-strain curves are shown in Supplementary Figure 7).”

Comment: *1. Figure 1D – the authors claim the partial restoration of the original crystal shape, I would like to see more results and analysis to support this conclusion. In particular, please add results using additional crystals and include analysis on the extent of recovery, does it depend on the extent of the initial mechanical bending? Can similar effect be observed without heating, i.e. can form II after bending transform to Form I at 300 K, and if yes, does that lead to change in the shape? What happens if other crystal shapes are studied, i.e. more prism-like or rods with different aspect ratios.*

Response to the comment: We thank the reviewer for this important remark. We have replace some of the panels in Figure 1 (panels B, C and D, and added panel F) with new images that show the same phenomena and we hope they illustrate better the point that we discuss in the text.

As we have explained in the revised text, form II crystals of terephthalic acid are only obtained as elongated parallelepipeds by hydrothermal crystallization. These straight crystals can be bent on their (010) face. As we also highlighted in the revised text, the degree of shape restoration does depend on the way the initial mechanical bending has been performed as well on the thickness of the crystal. Partial shape restoration to straight shape is observed during phase transition (above 350 K) to form I. However, in several cases we have observed complete shape restoration to a straight crystal. In the revised version, we have now included the data on four crystals that were bent to a different degree, and then heated for shape restoration during the phase transition to form I. The shape restoration of these bent form II crystals can only observed by heating and occurs above 350 K. The new examples were included in the revised version as Supplementary Figure 8. The following text was added in the main text of the manuscript:

“We noticed that among other factors—such as defects that are beyond the experimenter’s control—the ability for recovery of the straight shape is determined by the degree of bending and crystal thickness. In some cases, we were astonished to observe

complete recovery of the straight shape of the crystal (Supplementary Movies 8 and 9). Further, upon cooling to room temperature, form I crystals that have been straightened by heating are transformed to form II, and the initial bent shape of the crystals was recovered. These experiments demonstrate that the shape-memory effect occurs even on cooling; the crystals appear to retain ‘information’ of the deformation in their structure.”

Supplementary Figure 8 | Shape memory effects of crystals of form II terephthalic acid that were bent to various extent. The (initially straight) crystals of form II were mechanically bent to approximate angles of 20° (A), 30° (B), 55° (C) and 65° (D), as shown in the optical images. These bent crystals were then taken over the phase transition to form I by heating from 300 K to 362 K. Some of the thinner crystals having smaller bending angle showed complete shape restoration (panels A and B) while thicker

crystals having more acute bending angle showed partial shape restoration (panels C and D). Scale bars: (A) 500 μm , (B) 500 μm , (C) 400 μm , (D) 600 μm .

Comment: *2. What would happen if authors try to do these experiments multiple times by going back and forth between two polymorphs? For example, Fig 1D, last panel to the right, can form I crystal be converted to form II again? And if yes, can mechanical force now be applied to opposite direction so that mechanical bending is opposite to how it was done initially?*

Response to the comment: Following the reviewer's suggestion, we have performed additional experiments to verify this. A crystal of form II was bent by applying mechanical force on its (010) face. This bent crystal is then transformed to form I by heating above the phase transition temperature (354 K). During this phase transition the bent shape transformed to straight shape. Upon cooling to room temperature, form I crystal transformed to form II, and the shape of the crystal as again converted to the initial bent shape. This transformation between bent shape and straight shape can be repeated several times. We have now included a figure (Supplementary Figure 9) and a movie (Supplementary Movie 12) showing transformation between bent and straight shapes for two cycles. No deterioration of the crystal could be observed during these cycles. Mechanical bending of form II crystals can be performed on both faces (010/0 $\bar{1}$ 0). The following text was added in the main text:

"This transformation between bent and straight shape of the crystal can be repeated several times by alternative heating and cooling, and no deterioration of the crystal was observed (Supplementary Figure 9, Supplementary Movie 12). Mechanical bending of form II crystals can be performed at different locations simultaneously on both faces (010) and (0 0) to deform the crystal into an "S" shape, and the resulting crystal also exhibits shape-memory effect (Supplementary Figure 10, Supplementary Movie 13)."

Supplementary Figure 9 | A bent crystal of form II terephthalic acid, consecutively taken twice over the phase transition to form I. During the heating from 293 to 359 K the bent crystal recovered its straight shape. After cooling to room temperature it returned to its initial bent shape. The crystal does not show any visible deterioration compared to its initial appearance. Scale bar, 600 μm .

Comment: 3. Authors mention Young's modulus of form II, would be valuable to include result for form I as well. I also recommend adding hardness results here as well, in particular since significant plastic deformations take place as a result of applying mechanical force. Can change in the Young's modulus between two forms be part of the explanation for the observed shape change, for example difference in the mechanical properties of two forms changes stress/strain which is build up during the transformation, that could explain observed partial recovery. I think further experiments + corresponding discussion would strengthen the paper in addition to primarily structural arguments.

Response to the comment: We thank the reviewer for this suggestion. We performed nanoindentation measurements on both form II and form I and the corresponding graphs

are now included as Supplementary Figure 21 (shown below). The Young's modulus of form II is 6.2 ± 0.7 GPa and the hardness of the crystals is 0.33 ± 0.05 GPa. However, form I is comparatively softer, and its Young's modulus is 0.6 ± 0.2 GPa and hardness of the crystal is 0.05 ± 0.03 GPa. The difference in mechanical properties between the two forms could be the reason for form I being able to accommodate the local pressure in the bent region. The following text was added to the main text:

“As established by nanoindentation, the crystals of both polymorphs are generally soft (Supplementary Figures 7 and 21). The Young's modulus of form II is 6.2 ± 0.7 GPa and its hardness is 0.33 ± 0.05 GPa. Form I is comparatively softer in nature; its Young's modulus is 0.6 ± 0.2 GPa and its hardness is only 0.05 ± 0.03 GPa. The relatively softer nature of form I could compensate for the local pressure that must develop after bending on the concave side of the crystal, and could facilitate the coexistence of form I and form II in the respective concave and convex regions of the bent crystal. The critical strain for bending, based on several crystal samples, was found to be $2.5 \pm 0.2\%$; below this strain no phase transition occurs (Supplementary Figure 22). The minimal force required to induce the phase transition of form II to form I by bending of the crystal obtained from the force-displacement profile along with the critical strain and averaged over eight crystals was determined to be 92 ± 5 mN (an exemplary force-displacement and stress-strain curves are shown in Supplementary Figure 7).”

Supplementary Figure 21 | Load-displacement curves of terephthalic acid crystals obtained by nanoindentation on the (010) face of form II (A) and form I (B).

Supplementary Figure 22 | Stress-strain curves for TA crystals of form II bent by the three-point bending method. The critical strain was found to be $2.5 \pm 0.2\%$.

Comment. 4. Figure 5, almost no discussion in text/caption, I had difficult time following it, I suggest expanding text + corresponding figure caption or try to simplify it.

Response to the comment: We have now incorporated labels on each panel for clarity and we also included complete details describing each panel in the figure caption:

“Figure 5 | Mechanism of the phase transition during bending and shape-memory effect of a single crystal of terephthalic acid. (A) Transformation of the crystal between forms II and I, and relationship between the unit cells in the two phases. (a1) Transformation between the unit cell orientations of form II and form I viewed normal to the crystallographic faces (001) and (010). (a2) Images of the crystal in the two phases with face indices. (a3) Variation in the dimensions and shape of the unit cell, shown as different views. (B) Mechanism of transformation of the two-dimensional crystal lattice (cartoon) during the phase transition induced by application of local pressure. (b1) Crystal of form II and schematic of its lattice before bending. The straight crystal of form II can be bent by applying pressure on its (010) or (0 0) face. The lattice here is shown along the crystallographic [100] direction. (b2) Crystal and schematic of its lattice after bending. The bent crystal is a bimorph with two coexistent phases in the bent region separated by a phase boundary. (C) Mechanism of the shape-restorative effect. (c1) A bimorph in the kink of the bent crystal, shown with its lattice and unit cell. Unit cell orientations of both form II and form I in the bent region. (c2) Partial restoration of the straight shape of the crystal, and schematic representation of the underlying phase transition to form I”

Response to the comments from Reviewer #3

Comment: *Mechanically responsive molecular crystals have been a topic of significant interest in the last few years, with all sorts of surprising properties being observed (twisting, bending, elongation, jumping, etc). This really nice paper demonstrates that applying mechanical force to terephthalic acid crystals results in bending and partial phase transition to a different polymorph. However, heating reverses the phase transition and at least partially restores the original crystal shape, which is reminiscent of shape-memory polymers. While one can find other examples of reversible bending and mechanical responses induced by phase changes in the literature, I believe the combination of mechanically induced plastic bending that retains its shape which can then be restored thermally is pretty unique.*

The second key aspect that stands out in this work is that they were able to provide strong evidence for the proposed mechanism by obtaining crystal structures on the concave and convex sides of the bent crystal to demonstrate the partial phase transformation, and they argue convincingly how these changes in crystal packing relate to the rod deformation. The combination of the unique crystalline behaviors and the strong structural evidence makes this paper worthy of publication in Nature Comm. I have only a couple very minor comments:

Response to the comment: We thank the reviewer for the generally positive assessment of the work and the valuable suggestions. All suggestions were considered and addressed during the revision.

Comment: *- Is phase II of TA generally more stable under isotropic/hydrostatic pressure, or does the phase transition only result from the anisotropic nature of the local forces being applied here?*

Response to the comment: We confirmed, based on the reported literature on this compound, that form II of TA more stable under isotropic/hydrostatic pressure while form I is metastable. Moreover the application of anisotropic nature of the local forces causes phase transformation to form I. The following sentence was modified in the results section to reflect this (we cited the article by Roger Davey as only one of the articles that describes thoroughly the two forms and the phase transition that related them):

“Under specific crystallization conditions (slow evaporation of solutions or hydrothermal crystallization), TA can be obtained as two polymorphs, one of which is stable (form II), while the other is metastable (I) form at ambient conditions.⁴⁶ Crystals of both forms are exceptionally robust and can be converted into each other by heating or cooling without visible deterioration (Figure 1B, Supplementary Figure 1). The thermal profile of the phase

transition confirms that it is of first order⁴⁶ and occurs with a thermal hysteresis of about 33 K (Supplementary Figure 2)."

Comment: - *The paper notes large differences (1.80 vs 1.44 Ang) in the hydrogen bond lengths for the two polymorphs, but the crystal structure details for the COOH groups in the CIF files are suspect. In both polymorphs, the intermolecular O..O distances are very similar at 2.62 and 2.63 Angstroms. However, in one case, the O-H covalent bond is surprisingly short (0.82 Ang), and in the other it seems too long at 1.19 Ang. This then translates to the large differences seen in the H..O hydrogen bond lengths. I am not a crystallographer, and I recognize that H atom positions can be difficult to infer, but this still seems wrong (or at least something to discuss). None of this will likely change any of the important arguments, but they should look into the issue.*

Response to the comment: We thank the reviewer for noticing the long H...O hydrogen bond lengths, and we share the reviewer's opinion on the unreliable positioning of the hydrogen atoms with X-rays. We have now refined the hydrogen positions, using the common constraints, to acceptable distances. All data that pertain to the crystal structures have been corrected, and the new set of data are deposited with the revised version. The respective files in the CCDC were also replaced.

Response to the comments from Reviewer #4

Comment: *The authors report synchrotron microfocus X-ray diffraction analysis of the bent crystal of terephthalic acid (TA) that has two coexisting polymorphs and suggest a new mechanism for bent crystals as a mechanically induced phase transition. The introduced mechanism provides some insight on shape memory and restorative effects seen in other molecular crystals. However, many of the results discussed in the paper including the shape memory and self-healing effects of this crystal have already been discussed in their previous work (J. Am. Chem. Soc. 2016, 138, 13298). Specifically, most of the results presented in Figures 1~3 of this work have been discussed before in detail. There are numerous additional results that should be added overall and some of the arguments are not well supported. Thus this work requires major revision by addressing all questions and points suggested below before being considered for publication in Nature Communications.*

- Abstract and conclusion generalize the mechanism seen in TA crystal to soft crystals and organic solids, but this study only focuses on the TA crystal (which most of the results have been discussed in JACS 2016). There is not enough evidence to support that what happens in the TA crystals is generalizable to other molecular crystals.*

Response to the comment: We thank the reviewer for the apparently very thorough reading of the manuscript and the constructive suggestions. We took some time (about three months) to perform additional experiments and to respond to each of the comments, and we hope that we have provided a satisfactory response in the revised version. Extensive amount of new results have been added as supplementary figures and tables.

We do share the reviewer's sentiment about our preliminary results which were reported together with other results in our previous publication which focused on the mechanosalient effect. Indeed, we noticed some of the salient features described here in that stage of the research work, although at that time we were not able to fully explain our observations, which were also made on a very limited number of crystals. Some of these observations were reported at conference meetings, but remained in the form of mere observations. With this report, we intended to explore the full details of these effects, in an attempt to provide thorough experimental results and to explain the mechanism, as well as to reach more general conclusions. For instance, the partial recovery of the shape of the crystal or the shape-memory effect were noted in the earlier report, but the reasons for these observation remained unexplained because we did not have experimental facilities to pursue in that direction.

Having said that, the authors of this work strongly believe that experimental observations should be reported, regardless of whether they can be explained or not, because (as it has been the case in so many instances in science, and with an example most closely to our research – the thermosalient effect) they could become the seed or motivation for further in-depth studies, and in some cases, even lead to major discoveries. Perhaps the most important aspect of the new results presented in this work is that we have succeeded to obtain evidence for the first ever example of crystal deformation (bending) which is accompanied by a phase transition. This mechanism has not been reported before. We think it could be important because in principle the mechanism should be applicable to other crystals which have stable and metastable phases that can coexist at a certain temperature and pressure. We also provide evidence of coexistence of two different phases in the bent region, and we note again that this result has not been reported before. We hope that considering the depth and the breadth of the new results (in view of the different techniques used to characterize the effects) presented in this manuscript, the work can hopefully stand on its own, due to both the different focus of the main research theme, the methods used (SR XRD, SEM, optical microscopy, thermal data), the new mechanism, and also the other related properties that might be of a more general relevance and could potentially have a broader impact. We do, however, realize that additional results would contribute to strengthen further the originality and novelty aspect of this work.

In the revised version of the manuscript, and also because of the recommendation from another reviewer of this manuscript (see above), we have revised the abstract, to a smaller extent the introduction, and to a larger extent the conclusions sections, to reflect the possible general impact of the results presented in the manuscript, and in line with

the above applicability of the mechanism to other crystals. We believe that once the details of the mechanism become available to the research community, they could be useful to others to try and apply the mechanism in order to accomplish reversible plastic bending of organic crystals without delamination. The enhanced mechanical compliance in such materials would be the true benefit and the general relevance of the results presented here.

Comment: • *Phase identification in the bent region requires much more detail. This is the critical part of the paper. The data presented does not fully support the conclusion that the bent region is bimorphic. The microdiffraction results presented in Figure S5 is not of very high quality (showing doublets). Also the result is only presented in the compressed region, a single diffraction experiment, but not over the entire bent region as a mapping result. Mapping is needed to show how the polymorphs vary: is it continuous or an abrupt change from form I to II? The diffraction image presented needs to be indexed and the unit cell refinement results need to be presented with R values (Table S1 does not contain this info).*

Response to the comment. We thank the reviewer for this important comment, and we agree with them that the phase identification of the bent section is one of the central results in this manuscript.

During the work on this problem, we have of course, attempted mapping of the bent region, similar to the beautiful work on elastically bendable crystal published by McMurtrie Clegg and the collaborators recently (*Nat. Chem.* 10, 65–69 (2018)). After many trials, however, we concluded that similar analysis by mapping was not possible in our case, both because we have two distinct phases, as well as because of technical limitations (the limited beamtime we could obtain for these experiments). As the reviewer might be aware, these experiments are extremely challenging, and this is the reason none of these crystals have been characterized yet by X-ray diffraction, even when using bright X-rays and small X-ray beam, although the literature now contains a number of reports which show images of diffraction from bent crystals (see our recent minireview: P. Commins, D. P. Karothu, P. Naumov, “Is a bent crystal still a single crystal?”, *Angew. Chem. Int. Ed.* 2019, in press (<https://doi.org/10.1002/anie.201814387>)). Namely, with our experimental geometry at the synchrotron it was extremely difficult to design an appropriate data collection strategy in order to collect sufficient coverage for one of the forms in presence of the other form. Unlike the case where the structure changes continuously across the kink (i.e. the work off McMurtrie and Clegg), in case of a bimorph, the presence of two phases places limitations on the rotational freedom of the crystal because at different point of time the beam passes through one, second or both phases in different ratios, bot of them having spatially non-uniform orientation of their crystal lattice. The difficulties which are encountered with one phase are doubled when working with two phases. These technical issues are further complicated with the thickness of the crystal in respect to the limited diameter of the microfocus beam required to obtain sufficient count without damaging the crystal, restrictions with the phi rotation and of course, the strain on both lattices close to the habit plane between the two phases. Facing these difficulties, we had to resort to structure determination of the two domains representing different phases,

although this exercise also proved extremely challenging and required careful planning of the data collection strategy and technical experience with the data collection and processing. Inspection of the reciprocal lattice showed absence of twinning in the bent region. The initial inspection of the diffraction images and the rocking analysis farther from the habit plane that they were of sufficient quality, comparable to that of the straight section of the same crystal, and we collected full data on positions which were promising in view of the data quality. In the revised version, we have provided exemplary images of the diffraction collected at three sections of the crystal, the concave side, the convex side, and the straight section of the same crystal, with indices for several of the more characteristic reflections. They are now provided as Supplementary Figures 17, 18 and 19. Any deformations of any of the reflections in these images probably comes from the strain imposed on the two lattices by the habit plane between them. In the revised version of the supplementary material, we also provide complete data for the refined structures (Supplementary Table 4; the table was mislabeled in the original version, and we thank the reviewer for bringing that to our attention). As it can be inspected from the table, the crystal structures of both phases have been refined to good acceptable statistics. The same data were deposited and are available from the CCDC.

In addition to the X-ray diffraction data, we sought for other means to understand better the nature of the interface between the two forms. Optical micrographs, and especially, the scanning electron micrographs, shown in Figure 3A and Supplementary Figure 16 were particularly helpful to identify the striations on the surface of the crystal and to determine their orientation for correlation with the crystal axes, and these were applied to a number of crystals to confirm the formation of two phases in the bent region. More importantly they helped to directly visualize the habit plane and the slight difference in striations on both phases that are joined at the plane. For bent crystals, we provided additional evidence by using in-house diffractometer (the data quality was only sufficient to confirm the unit cells) and Raman spectroscopy. We rely on these additional experimental results, in addition the SR X-ray diffraction data as the central evidence for the mechanism of bending that we advance in this work.

Comment: *Section “Phase transition and shape restorative effects”*

• *Timescale on transition: In the first paragraph, when they indicate that the crystals “do not transition sharply”, “rapid reshaping upon mechanical stimulation”, etc. what does that mean? Is it 1st order or 2nd order transition? Need to show DSC results which show 1st order transition. On a related note, Supplementary Videos do not have time stamps or scale bars*

Response to the comment: We considered that the general information on the nature of the phase transition is available from the literature, and this is why we did not provide more information in the original version of the manuscript. In the revised version, we included a new figure in the supplementary material (Supplementary Figure 2) which contains detail thermal analysis, by DSC, of the transition between the two forms (for convenience, the figure is also shown below) after one thermal cycle and after four thermal cycles, without grinding and with light grinding of the sample (this was necessary,

because from our prior experience we were aware that grinding can have a substantial effect on the thermal profile of thermosalient transitions).

Ass can be inspected from Supplementary Figure 2, the transition from form II to form I shows a relatively large thermal hysteresis of ca. 33 K and the peaks confirm that the transition is of first order. This result is consistent with the previous literature that has investigated the same phase transition, where it was characterized as being of first order (for example, R. J. Davey, S. J. Maginn, S. J. Andrews, S. N. Black, A. M. Buckley, D. Cottier, P. Dempsey, R. Plowman, J. E. Rout, D. R. Stanley, A. Taylor, Morphology and Polymorphism in Molecular Crystals: Terephthalic Acid, *J. Chem. Soc., Faraday Trans.*, 1994, 90, 1003). The order of the transition is also consistent with thermosalient transitions of other thermosalient solids, all of which are without exception of first order (*Chem. Comm.*, 2016, 52, 13941; *Chem. Rev.* 2015, 115, 12440). We also would like to note again that the DSC of thermosalient crystals are usually recorded from non-powdered samples because during powdering of these soft solids, usually the thermal effect is either alleviated or completely suppressed (*J. Am. Chem. Soc.* 2013, 135, 12241). This is the reason why in the new Supplementary Figure 2 we have also included DSC patterns of crystals that were lightly powdered. The two sets of figures (crystals and powder) show the effect of grinding on the appearance of the DSC profiles of the thermosalient solids (as highlighted in *J. Am. Chem. Soc.* 2013, 135, 12241). We also thank the reviewer for their comment on the time stamps and the scale bars. We have now added time stamps and scale bars in all supplementary videos.

The following sentence was added in the main text, and the above reference was cited (ref. 46): “*The thermal profile of the phase transition confirms that it is of first order⁴⁶ and occurs with a thermal hysteresis of about 33 K (Supplementary Figure 2).*”

Supplementary Figure 2 | Differential Scanning Calorimetry (DSC) of as-obtained (A, B) and lightly grinded (C, D) crystals of form II terephthalic acid (TA). (A) DSC profile recorded by heating and cooling of crystals of TA over the temperature of phase transition. (B) DSC profile recorded over four consecutive thermal cycles. Note the slight offset in the range of transition temperatures between the consecutive cycles. (C) DSC profile of lightly grinded crystals of TA. Note that similar to other thermosalient transitions (J. Am. Chem. Soc. 2013, 135, 12241), the peak intensity is significantly alleviated compared to the non-grinded crystals. (D) Two thermal cycles in the DSC of lightly grinded crystals. The heating and cooling rates in all experiments were 10 K min^{-1} .

Comment: • “Twinned crystals” can indicate that the two forms are twins, indicating that they are of same polymorph. In this work, they are form I and II, and are different polymorphs

Response to the comment: We agree with the reviewer on the importance of the accurate use of the term “twinning”. The term “twinning” is used to refer to two crystal components that have grown together. In the more exact formulation of twinning, which is given by a twin law for a particular twinned crystal, a twinned crystal is an aggregate in

which different domains are joined together and are related to each other with a specific symmetry operation. The diffraction patterns derived from different domains are rotated, reflected or inverted with respect to each other, depending on the symmetry relationship between the different domains. The diffraction pattern measured during complete data collection (which includes also phi rotation) is a superposition of all of these. In the case of the form II crystal of TA, the diffraction data were collected at mainly two locations along the crystallographic b axis by restraining the phi and 2θ angles. This excludes the overlapping of diffraction spots with the other form in the convex or in the concave region. We would like to point out that within the convex or concave regions the twinning may indeed occur, however we have evaluated the diffraction data of the convex and concave regions separately for twinning, and concluded that the region is devoid of twinning. In line with this, we have removed the term “twinning” in the text where it specifically referred to a bent crystal of TA.

Comment: • *From Supplementary Movie 1, it is hard to believe that the crystal springs off due to phase transition rather than force of the sharp object pushing down on the edge of the crystal. In the video the partial transition from I to II already takes place in the beginning before jumping off. Is there proof that the crystal completely transitioned to form II? Also, this phenomenon was already discussed in JACS 2016, it is unclear what the purpose of mentioning it again here is.*

Response to the comment: The thermodynamically stable polymorph, form II, is converted to form I by heating. Upon cooling of the thus obtained form I, most of the crystals are transformed back to the (more stable) form II. However, some crystals remain untransformed and are “trapped” as form I, which is metastable at room temperature. We have repeatedly observed that the transformation of these crystals can be induced by mechanical stimulation, i.e., by applying a local pressure, for example, by lightly touching the crystals with a hard object. Upon mechanical stimulation, they are rapidly transformed to form II. The rapid progression of the phase front results in simultaneous reshaping of the crystal and hopping off the stage, a phenomenon which was termed “mechanosalient effect”. In the revised version, we have included a few additional videos as supplementary material (Supplementary Movies 1–5) which clearly illustrate this effect with multiple crystals.

We would like to clarify the content of the original Supplementary Video 1. The original version of the video showed a crystal of form I, obtained as described in the preceding paragraph, which was lightly pressed at the edge, occasionally slightly sliding along the surface. In the first portion of the video (time 00:00 to about time 01:30 s), the crystal of form I was stepwise and partially converted to form II (the converted domain is visible and can be inspected from the shifting of the “kink” on the edge of the crystal, which represents the end point of the phase boundary between the two phases). If the crystal has defects, the transformation starts and is terminated at a defect, and therefore only one domain is converted at a time, and the entire crystal is converted over a longer timespan. As it can be seen from the original video, the first domain is transformed around 00:08-00:12 s. The

second domain is transformed around 00:28 s (note that these are high speed recordings, so what appears as slow transformation actually occurs very fast). Further application of pressure on the crystal and against the base (the needle visible in the video is towards the viewer pointing in the viewer's direction) by applying very light pressure on this transformed region) results in (indirect) pressure on the remaining part of the crystal, which then at 01:34 transforms completely. This transformation is fast and results in jumping of the crystal.

Because in this experiment the crystal was touched always on one side (from the top and close to the edge), in the revised version of the manuscript we have added four new videos (the new Supplementary Videos 2–5) which show this phenomenon more clearly. In these videos, the crystal is not pressed from the top or at the edge, but instead it is lightly tapped from the side, and therefore it does not experience any pressure applied from the top/edge that would appear as a reason for the crystal motion simply as a reaction to the action. Moreover, the high-speed recordings clearly show that the crystal springs off the support at a rate which is much faster than the rate by which it is approached from the side, and confirms the mechanosalient effect.

We agree with the reviewer that this phenomenon has been discussed in our previous work, however we felt that a combined narrative on the polymorphism, bending, and the mechanosalient effect, and a more detailed study of the latter that we provide here may shed further light on the mechanism, especially in view of the fact that the phenomenon is rooted in a phase transition which also is the reason for the new mechanism of crystal bending that we describe in this work. Presenting all these exotic phenomena together appeared to us to add to the completeness of this report. However, considering the preceding study, the mechanosalient effect was intentionally given much less attention and space in the manuscript relative to the bending, shape-memory and shape-restorative effects. We hope that by presenting another aspect of the same phase transition, this portion of the results adds to the completeness of the narrative in the overall manuscript. In addition to the new videos, in the revised version of the manuscript, we have included a new figure (Supplementary Figure 3) that illustrates the phenomenon by a set of examples of the mechanosalient behavior of form I crystals having different size, and we shifted the panel describing the phenomenon to panel F in Figure 1. Following the reviewer's suggestion, in the revised version, we have also included single crystal X-ray diffraction measurements of the unit cell performed on partially and completely transformed crystal at different locations (Supplementary Figure 4, Supplementary Table 1). We also recorded Raman spectra that confirm the phase identity of the two phases before and after mechanical stimulation (Supplementary Figure 5). These results confirmed that after the transformation induced by touch, the crystal was in form II (see the response to the following comment).

Supplementary Figure 3 | *Mechanosaltation effect of crystals of form I terephthalic acid captured by high-speed camera at a rate of 1500 frames per second. Initially, crystals of form II were heated and transformed to form I. After cooling, most of the crystals converted to form II, however some of the crystals remained in form I, which is metastable at room temperature and ambient pressure. These crystals jump when they are lightly contacted with a metal needle. Scale bars: (A) 600 μm ; (B) 500 μm ; (C) 400 μm ; (D) 300 μm .*

Comment: • *Figure 1B: Is there structural proof that the region that the sharp object touches is Form II? Was micro X-ray also done in that region? If so, it should be discussed, if not, there should be supporting evidence that the transitioned region is indeed the stable Form II, using other methods to cross validate such as spectroscopy.*

Response to the comment: As we discussed in our response to some of the previous comments, only the crystal of form I, which is metastable at room temperature, can be transformed to form II by applying local pressure (mechanosaltation effect). The original figure 1B may have not been very descriptive because the portion of the crystal that was attached to the base was in form II (converted upon affixation of the crystal), while the remaining, larger portion of the crystal was in form I. The section that was in form I was actually contacted with a metal object.

In order to clarify this aspect of the transformation further, in the revised version we have replaced the series of snapshots in Figure 1B with another series of images recorded

by using high-speed camera (Figure 1F). We also performed analysis of the mechanosalient effect in a crystal before, during and after the phase transition by determining the unit cell in all cases. This proved to be challenging because once the transition is initiated, it proceeds in very short intervals, even with lightest mechanical stimulation; it is thus very difficult to “trap” the crystal in a stage where only portion has been converted.

To that end, a single crystal mounted on the diffractometer head was first heated over the transition from form II to form I several times, and then slowly cooled to room temperature. At room temperature, form I is metastable. Once this metastable form I is contacted with a metal needle, it starts transforming to form II. This partially transformed crystal was covered in Paratone to slow down the conversion to form II, and the unit cell was determined with an in-house diffractometer at different locations to confirm the phase identity. After 15 min, the crystal was completely transformed to form II and single crystal diffraction measurements were performed again to confirm that the crystal is completely converted to form II. The locations where the unit cell was determined are marked in a new figure, which is now included in the revised version as Supplementary Figure 4. The crystallographic data are provided as new Supplementary Table 1. The identity of the phase in the two regions was additionally confirmed by using Raman spectroscopy. The Raman spectra are provided as Supplementary Figure 5.

Supplementary Figure 4 | *A single crystal used for in situ X-ray diffraction analysis of a mechanically stimulated phase transition. A crystal of form II terephthalic acid was covered in Paratone to slow down the transition and mounted on the diffractometer head. The crystal was taken over the transition between form II and form I several times by repeated heating-cooling cycles, and was slowly cooled to room temperature, where form I is metastable. Once the crystal in the metastable form I is lightly contacted with a metal needle, it starts to transform to form II. The unit cell was determined at different positions of the crystal before and during the transition to confirm its phase identity. After 15 minutes, the crystal was completely transformed to form II, and diffraction data were collected again. The unit cell parameters are available from Supplementary Table 1.*

Supplementary Table 1. Unit cell parameters determined for a crystal that was contacted with a metal object (labels A–D correspond to the respective panels in Supplementary Figure 4)

	A	B	C		D
Temperature / K	290	290	290	290	290
Polymorph	Form II	Form I	Form II	Form I	Form II
Crystal system	Triclinic	Triclinic	Triclinic	Triclinic	Triclinic
Space group	$P\bar{1}$	$P\bar{1}$	$P\bar{1}$	$P\bar{1}$	$P\bar{1}$
$a / \text{\AA}$	5.14	3.73	5.13	3.82	5.13
$b / \text{\AA}$	5.52	6.49	5.39	6.54	5.39
$c / \text{\AA}$	7.22	7.39	7.02	7.40	7.02
$\alpha / ^\circ$	71.92	82.95	72.35	83.40	72.35
$\beta / ^\circ$	75.84	81.49	76.91	80.78	76.91
$\gamma / ^\circ$	87.18	89.23	87.82	89.43	87.82
Volume / \AA^3	189	175	180	181	180

Supplementary Figure 5. Raman spectroscopic measurements. By touching with metal object form I crystal transform to partially converted form II crystal (A). Raman spectra (different domains of form I and form II) measured on different domains of form I and form II (B). Note: Here partially converted form II crystal is actually a crystal with different domains of form I and form II.

Comment: • *Figure 1D: This is identical to Figure 4A of JACS 2016, where they bend the crystal and heat. It is unclear what new information this figure adds to this work. Also, what happens to the crystal after cooling? Would the crystal transform back to Form II? Would you able to start from Form I and induce mechanical bending as well or does bending only occur on Form II? Need more details.*

Response to the comment: We share the reviewer's sentiment regarding the possible conceptual overlap with Figure 4A in our previous publication, at least in view of the visual presentation of the phenomenon in Figure 1D, although the two figures show different samples. As we have elaborated earlier in this response, in our previous publication the phenomenon was noted, but its mechanism remained speculative and it was not substantiated by experimental evidence. Specifically, we were not able to explain the mechanism of what appeared to be a shape recovery similar to that encountered with mesophasic materials, such as polymers; nevertheless we decided to report our observations. The authors of this work strongly believe in the importance of reporting observation, regardless whether they can be explained at the time when they were made. We also believe that the relevance of a follow-up, in-depth research should not be alleviated by avoiding to report such observations.

The scope of the present work was to provide an in-depth insight into the phenomenon and to investigate the details of this unusual property by using different analytical techniques. For completeness of the current study, we needed to show one panel with this phenomenon, so as to avoid the necessity for reference to a figure in an earlier publication. Specifically, in the work presented here we provide exhaustive experimental evidence with results that explain and provide evidence for our earlier observations. Moreover, by approaching the preliminary results with a further in-depth study we not only provide direct evidence for the structure of a crystal that can be bent without delamination, but we also establish the mechanism by which the crystal of TA bends as a new mechanism for bending of organic crystals, which could be of more general importance than the single example studied here. Namely, as we have also noted in the manuscript, this mechanism for crystal bending could be applicable (and indeed, practically more relevant) to other crystals which have metastable phases at ambient conditions. These conclusions, we believe, should warrant publication of the new results presented here.

In order to provide these new details on the phenomenon, Figure 1D was now replaced with another series of images which show a new property that we believe has an added value to the property that we observed before – the reversibility of the shape change that can be induced mechanically once and reverted thermally multiple times. We also show, both in the new Figure 1D and, due to space limitations to a greater extent in Supplementary Figures 8, 9 and 10, the behavior of the crystals of TA after single or repeated alternative treatment with heat and/or mechanical force. We hope that this novel aspect is acceptable to be presented together with the other results on this material.

Comment: • *Figure 1E: Supplementary Video shows clear restoration, but from the figure it is hard to see. Also, need to show evidence that after heating, the crystal is fully transformed to Form I. Was X-ray repeated on the crystal after heating? Since it did not completely restore is it possible that it may not be completely Form I? Heating alone can't guarantee that the crystal has completely transformed to Form I.*

Response to the comment: We have included with this revision new videos (Supplementary Movies 14–19), figures (Supplementary Figures 11 and 13) and tables (Supplementary Tables 2 and 3) that show images and contain data related to the restoration of crystal during heating (the tables contain the unit cell parameters). When the crystal of form II is heated to 359 K, it is completely transformed to form I. In order to confirm that, we have performed single crystal X-ray diffraction analysis on different locations of such crystal. Other crystals were damaged and we recorded X-ray powder diffraction above 359 K. the powder diffraction pattern in the high temperature phase corresponds to that of form I calculated from its crystal structure. These results confirmed that in all cases the samples were fully converted to form I after heating. The corresponding powder diffraction and single crystal X-ray diffraction results are now included in the Supplementary Material (new Supplementary Figure 15). We would like to retain Figure 1E in the main text because we consider that it depicts well the partial shape-restoration on a heavily damaged crystal. For convenience, the Supplementary Figures 11-15 showing images of damaged crystals and locations where the diffraction data were collected, as well as the new Supplementary Tables 2 and 3, are provided below.

Supplementary Figure 11 | Mechanism of restoration by heating of the integrity of cracked crystals of form II TA which were pressed uniformly on their (001) face. (A) Schematic of the restoration mechanism. (B,C) Shape restoration of lightly compressed (B) and heavily compressed (C) crystals. Scale bars: (B) 300 μm ; (C) 800 μm .

Supplementary Figure 12 | Image of a crystal of TA used in the unit cell measurements of the self-restoration effect of pressed crystals. Initially, a crystal of form II was damaged by applying force with a metal plate on its (001) face and mounted on the loop. The damaged crystal was taken over the phase transition temperature and its unit cell was determined at four locations (marked a, b, c and d). The unit cell corresponds to that of form I. The unit cell details are available from Supplementary Table 2.

Supplementary Figure 13 | Propensity for restoration, after heating, of integrity of crystals of form II TA that were damaged with a metal object on their (010) face. (A) Schematic of the shape restoration mechanism of a crystal that has been damaged by applying relatively uniform pressure across its surface by using a metal object. (B,C) Shape restoration of slightly (B) and heavily (C) damaged crystals. Scale bars: (B) 200 μm ; (C) 700 μm .

Supplementary Figure 14 | Image of a crystal of TA used in the unit cell measurements of the self-restoration effect. Initially, a crystal of form II was damaged by applying pressure on its (010) face with a metal object. The damaged crystal was taken over the phase transition temperature and its unit cell was determined at three locations (marked as a, b and c). The unit cell corresponds to that of form I. The unit cell details are available from Supplementary Table 3.

Supplementary Figure 15 | Identification of the high-temperature phase of terephthalic acid by using powder X-ray diffraction. Comparison between the experimental powder X-ray diffraction patterns of form II (A) and form I (B) with the pattern of form I calculated from the crystal structure (C).

Supplementary Table 2. Unit cell parameters determined for a shape-restored crystal after it has been uniformly pressed with a metal plate (the locations a–d refer to Supplementary Figure 12)

	Damaged form II	a	b	c	d
Temperature / K	290	362	362	362	362
Polymorph	Form II	Form I	Form I	Form I	Form I
Crystal system	Triclinic	Triclinic	Triclinic	Triclinic	Triclinic
Space group	$P\bar{1}$	$P\bar{1}$	$P\bar{1}$	$P\bar{1}$	$P\bar{1}$
a / Å	5.02	3.77	3.70	3.72	3.72
b / Å	5.34	6.47	6.49	6.46	6.48
c / Å	6.99	7.38	7.33	7.38	7.36
α / °	72.01	82.94	83.85	83.31	83.35
β / °	76.01	81.55	80.37	80.45	80.03
γ / °	87.32	88.99	86.88	88.54	87.50
Volume / Å³	173	177	173	174	174

Supplementary Table 3. Unit cell parameters determined for a shape-restored crystal after it has been heavily damaged by using a metal object (the locations a–c refer to Supplementary Figure 14)

	Heavily damaged form II crystal	a	b	c
Temperature / K	290	360	360	360
Polymorph	Form II	Form I	Form I	Form I
Crystal system	Triclinic	Triclinic	Triclinic	Triclinic
Space group	$P\bar{1}$	$P\bar{1}$	$P\bar{1}$	$P\bar{1}$
a / Å	4.97	3.93	3.80	3.81
b / Å	5.37	6.36	6.39	6.42
c / Å	6.94	7.35	7.40	7.38
α / °	72.02	84.58	83.42	83.05
β / °	75.85	79.65	80.89	81.08
γ / °	85.90	88.04	88.28	88.42
Volume / Å³	171	180	176	177

Comment: • *Figure 2: Again, how is it proven that it is entirely Form I at 358K? The bottom figures and the supplementary movie seems to show that maybe the crystal has partially transformed from Form II to I, as the phase propagation does not seem to go through all the way. Does it eventually transform completely? Is phase transformation affected by defect density or location of defects caused by the metal plate?*

Response to the comment: The crystal shown in Figure 2 was mechanically damaged by compressing it uniformly with a metal plate. During this procedure, a kink developed along the long axis of the crystal. Careful examination of the supplementary movie 17 shows that upon heating to the phase transition temperature, the shape restoration process propagates starting from both ends of the crystal and is eventually completed at the kink. In the revised version, we have included Supplementary Figures (see above) and Supplementary Movies of several crystals that show complete phase propagation during heating. We have also performed single crystal X-ray diffraction on the cracked crystals of TA before and after the phase transition. These results confirm that the transformation is complete (the corresponding data are now included in the Supplementary Material, and are also provided below for convenience). The phase transformation of form II to form I is partially affected by the defects caused by the metal plate. During heating, the propagation is terminated at defects and the transformation of a larger portion of the crystal is delayed for a few milliseconds. However, the transformation is eventually completed once the temperature is raised above 358 K.

Comment: *Section “Microfocus X-ray diffraction analysis of a bent crystal”*
• *As indicated in manuscript, Supplementary Figure 5 shows long range order, but also shows several diffraction peaks close together, which can indicate twinning or gliding of the different layers. Explanation of what they are is needed.*

Response to the comment: We have carefully evaluated the synchrotron X-ray diffraction data collected from the bent region (both on the convex and the concave side of the kink) for twinning, and the data was also checked against alternative unit cell parameters. The inspection for possible twinning and the reciprocal lattice view of form I and form II in the bent region did not show any twinning. However, a few peaks appear close together and we believe that this is due to the compression of the layers during the bending. We have included more diffraction images in the revised supplementary material (Supplementary Figures 17–19 in the revised version) that show that almost all reflections are single reflections. The figures are provided below for convenience.

Supplementary Figure 17 | Examples of diffraction images recorded from the concave (inward) side of the bent region of a TA crystal using microfocus X-ray diffraction with synchrotron radiation. The location where the diffraction was recorded is marked with an arrow in panel A, and exemplary diffraction images are shown in panel B.

Supplementary Figure 18 | Examples of diffraction images recorded from the convex (outward) side of the bent region of terephthalic acid crystal using microfocus X-ray diffraction with synchrotron radiation. The location where the diffraction was recorded is marked with an arrow in panel A and exemplary diffraction images are shown in panel B.

Supplementary Figure 19 | Examples of diffraction images recorded from the straight part of a bent crystal of terephthalic acid using microfocus X-ray diffraction with synchrotron radiation. The location where the diffraction was recorded is marked with an arrow in panel A, and exemplary diffraction images are shown in panel B.

Comment: • *The manuscript describes Supplementary Table 1 as the result of refinement on the bent convex and concave side crystals, but the title of the table is: Crystallographic data and refinement details of terephthalic acid structures refined from the same crystal at different temperatures. It seems like the title is labeled incorrectly.*

Response to the comment: We thank the reviewer for catching up this error. Now the labels in the Supplementary Table 1 (Supplementary Table 4 in the revised version) corresponding to convex and concave region of bent crystal are corrected.

Comment: • *It would be interesting to see if there is a critical strain for forming I on the compressed concave side of the crystal. With less than critical strain, then both sides of the crystal even after bending may still be Form II.*

Response to the comment: In order to verify this, we bent several crystals using three-point bending method and the critical strain was found to be $2.5 \pm 0.2\%$. We did not observe any phase transition below the critical strain. The following sentence was added to the main text:

“The critical strain for bending, based on several crystal samples, was found to be $2.5 \pm 0.2\%$; below this strain no phase transition occurs (Supplementary Figure 22). The minimal force required to induce the phase transition of form II to form I by bending of the crystal obtained from the force-displacement profile along with the critical strain and averaged over eight crystals was determined to be 92 ± 5 mN (an exemplary force-displacement and stress-strain curves are shown in Supplementary Figure 7).”

Supplementary Figure 22 | *Stress-strain curves for TA crystals of form II bent by the three-point bending method. The critical strain was found to be $2.5 \pm 0.2\%$.*

Comment: • *It would be interesting to see if the crystal integrity can still be maintained if the crystal is bent the opposite way.*

Response to the comment: Form II crystals of TA can be bent either by exerting pressure either on their (010) face or on their (0 $\bar{1}$ 0) face. Our experiments showed that the crystal integrity is well preserved in both cases. We have included in the revised version of the supplementary information a new figure (Supplementary Figure 10) that shows the mechanical bending by applying force on both faces. Consistent with the other results, partial shape recovery was observed when this crystal was heated above the phase transition temperature. Upon cooling the crystal regains its original bent shape on both faces.

Supplementary Figure 10 | Deformation of a crystal of form II terephthalic acid upon application of force on opposite faces of the crystal. (A) Schematic of the three-point bending of the crystal performed by applying force on its (010) and (0 $\bar{1}$ 0) faces. (B) Shape-memory behavior of an S-shaped crystal upon heating and cooling. Scale bar: 800 μm .

Comment: • *A comparison of the bent crystal Form II to the pristine Form II at same temperatures would be interesting. Since the convex part still experiences expansion as Form II, does it have expanded/contracted unit cell values when compared to the pristine case?*

Response to the comment: We thank the reviewer for this suggestion. We have determined the unit cell of two different regions of the same bent crystal, and the results are included as Supplementary Figure 23 and Supplementary Table 5. showing unit cell parameters of bent crystal Form II and pristine Form II at same temperature.

Supplementary Figure 23 | Image of the bent form II crystal of TA used for unit cell measurements on both bent and straight parts. The unit cell details are available from Supplementary Table 5.

Supplementary Table 5. Unit cell parameters determined from the straight and bent sections from the same bent crystal of form II terephthalic acid

	Bent form II	Straight form II
Temperature / K	290	290
Polymorph	Form II	Form II
Crystal system	Triclinic	Triclinic
Space group	$P\bar{1}$	$P\bar{1}$
$a / \text{Å}$	5.03	5.11
$b / \text{Å}$	5.38	5.33
$c / \text{Å}$	7.00	6.99
$\alpha / ^\circ$	72.28	72.29
$\beta / ^\circ$	75.84	76.04
$\gamma / ^\circ$	87.14	88.11
Volume / Å³	175	176

Comment: Section “Mechanism of plastic bending by partial phase transition”
• Authors described in last paragraph before Figure 5 “This causes sliding and rotation of nearly 30° of the tapes on the concave side, ...” which angle is this referring to? The author never described in the manuscript.

Response to the comment: In the revised version we modified the sentence to clarify its contents. In form II, due to π — π stacking, the molecular tapes form infinite stacks along the crystallographic *a* axis. Furthermore, the molecular tapes are connected via C—H...O contacts along the crystallographic *c* axis. The molecular tapes in form II are stacked at a tilt angle $\sim 55^\circ$ with respect to the crystallographic *a* axis. During the phase transition this tilt angle changes to $\sim 83^\circ$, and the molecular tapes slide and are realigned to new positions.

The following sentence was modified:

*“This causes sliding and rotation of the molecular tapes in form II to nearly 30° with respect to the crystallographic *a* axis on the concave side, where they reform the sheets along the crystallographic *c* axis, and the concave part of the crystal is transformed to form I.”*

Comment: • Also authors mentioned in the same paragraph that: “The slight mismatch of the two unit cells, which—similar to their coexistence as alternating layers in a straight partially converted crystal—remain attached at the phase boundary, stabilizes the bimorph and the bent shape of the crystal is retained (Figure 4D).” As authors show the lattice structure in Figure 5A the mismatch at the phase boundary looks significant (for instance around 26.3 % decrease in *a*-axis length); which do not mean slight mismatch of the two unit cells. And in Figure 5B on the right schematic exaggerated the lattice dimension of Phase I to seem like it matches to that of Phase II.

Response to the comment: We have now changed the sentence accordingly in the revised version and also removed the exaggerated lattice dimension of phase I in Figure 5B:

“The mismatch of the two unit cells, which—similar to their coexistence as alternating layers in a straight partially converted crystal—remain attached at the phase boundary, stabilizes the bimorph and the bent shape of the crystal is retained (Figure 4D).”

Comment: • *Incorrect statement right before Figure 5: “When the bent crystal is heated over the phase transition, the form I on the concave side is transformed back to form II” (it should be form II to form I)*

Response to the comment: We have rechecked the mentioned sentence and it is indeed correct. In the bent crystal, the convex side is in form II and the concave side is form I. When the bent crystal is heated over the phase transition, the form I on the concave side is transformed to form II.

Reviewers' comments:

Reviewer #1 (Remarks to the Author):

The authors have clearly disagree with my previous comments and they have not been addressed - the authors have not really made an attempt to understand them, even when they parallel the comments of other referees. The responses to my comments are not endearing and are not really appropriate - particularly the discussion of "what is a crystal" to which the response is - we have published a mini-review elsewhere (which contains significant opinion) and the conclusions of our other opinion are that we are correct!

Having noted and read the extensive, but too long and unconvincing responses to the referees I have reread the paper fresh to consider it in the absence of my previous comments - ie as if it were a fresh submission.

Overall this paper contains some careful experimental work which investigates the well-known terephthalic acid system and tries to understand the changes that take place upon bending a crystal plastically. It has also been observed that after plastic deformation the crystals can partially regain their shapes upon a number of heating and cooling cycles. I am concerned that the findings, however, while interesting are not really supported by the evidence - the conclusions are essentially hypotheses and nothing has been done to establish whether or not the hypotheses hold. The authors have attempted to generalise their results with broad statements (even the title), without any evidence of the generalities. The mechanism for bending is conjecture and there is not actual evidence to support the conclusions drawn, particularly with reference to a rotation of the molecules in the crystal. I think this paper would be acceptable if the quality of scholarship were improved - but at the moment it is more conjecture and opinion than science and fact.

1. Line 52 - this statement is incorrect. Plastic bending results in the loss of discrete diffraction - see Thomas et al.

2. Line 64-68 the authors assert a new mechanism of bending. This is not supported - it appears that the mechanism of bending results from slippage of molecules (which is accompanied by a phase change), not a new mechanism. This is also a chicken and egg question.

3. There are not statistics or errors or objective measures the restorative effects. It is likely that the origins of these effects are defects and/or phase boundaries introduced upon deformations

4. There are many statements like "it develops cracks on its surface ... but the slabs remain conjoined" - there is no evidence for this other than a photo - how do the authors know the cracks are only on the surface!!!?

5. In figure 2 a mechanism of "apparent" restoration is proposed - this suggests that there is no restoration, it just appears to be restored!

6. The authors use microfocus diffraction to analyze the inside and outside of a bent crystal and determine that two phase are present. By definition the resulting biphasic material is no longer a crystal - but that is of little consequence. The authors are unable to face-index these positions and as the crystals are triclinic the settings are arbitrary. Therefore there can be no conclusion drawn about θ vs θ' . The changes in molecules are important not the changes in unit cells.

7. It is much more likely that the reason for bending is planes of molecules slipping past each other, as noted by the authors between lines 270-306. Indeed inspection of the two crystal structures of the two polymorphs shows that the main difference between the two is a slippage of 0.7 Å along a chain in polymorph 2 compared to polymorph 1. All other distances (ie separations between chains etc) remains identical. This slippage has been observed by the authors by their experiments. This slippage will be responsible for the bending - slippage is the mechanism - this is a known mechanism, not a new mechanism.

8. The authors haven't discussed the similar results of terephthalamide - a very similar molecule,

with similar crystal packing. In this case application of stress results in a phase change and bending. Removal of the force results in the reversal of the phase change and restoration of the original shape. The only potential differences are: 1. lack of hysteresis 2. elastic vs plastic behaviour. The authors need to discuss this in detail and propose why one is elastic and one is plastic for there to be any real significant value here. It is disingenuous to ignore this previous careful study.

At present I don't believe this paper is suitable for publication, because of the low level of scholarship/overhyped claims. If it were reworked I think it would be a valuable contribution.

Reviewer #2 (Remarks to the Author):

I carefully read the responses from the authors to my comments/suggestions as well as those raised from other reviewers. In my opinion, the authors did an amazing job and fully addressed my concerns and incorporated my suggestions, including challenging mechanical characterization, the paper is much more refined now, a general public appeal is much clearer and I recommend accepting it for publication at Nature Comm in the present form.

Reviewer #3 (Remarks to the Author):

The authors have addressed my concerns in the revision, and I support publication of the manuscript in its current form.

Reviewer #4 (Remarks to the Author):

The authors have made significant effort to thoroughly address the concerns from the manuscript.

Two questions that still arise from the response are regarding these responses:

1) Original Comment: It would be interesting to see if there is a critical strain for forming I on the compressed concave side of the crystal. With less than critical strain, then both sides of the crystal even after bending may still be Form II.

Response to the comment: In order to verify this, we bent several crystals using three-point bending method and the critical strain was found to be $2.5 \pm 0.2\%$. We did not observe any phase transition below the critical strain.

New comment: For Figure S22, is the plateau that is observed definitively related to the Form II to I transition? Is it possible that it can be from gliding of the planes?

2) Original Comment: Also authors mentioned in the same paragraph that: "The slight mismatch of the two unit cells, which—similar to their coexistence as alternating layers in a straight partially converted crystal—remain attached at the phase boundary, stabilizes the bimorph and the bent shape of the crystal is retained (Figure 4D)."

As authors show the lattice structure in Figure 5A the mismatch at the phase boundary looks significant (for instance around 26.3 % decrease in a-axis length); which do not mean slight mismatch of the two unit cells. And in Figure 5B on the right schematic exaggerated the lattice dimension of Phase I to seem like it matches to that of Phase II.

Response to the comment: We have now changed the sentence accordingly in the revised version and also removed the exaggerated lattice dimension of phase I in Figure 5B

New Comment: Our concern of the mismatch of the two unit cells and especially the 26.3%

difference in the a axis is still not well addressed. Specifically, in figure 5B, even in the revised figure, it does not accurately show the 26% difference in the a axis, which is significant. It seems unlikely that they would be sharing a common phase boundary.

Response to the comments from the reviewers

We thank all reviewers for the valuable comments which have contributed significantly to improve the quality of our manuscript. We considered all comments, and we tried to address them to the best of our ability. Unless stated otherwise, all numbers of the figures and supplementary materials refer to the *revised* version of the manuscript. Together with this submission we have provided marked copies of the main text and the supplementary materials where the changes to the original version have been marked. For convenience, in what follows, the original comments from the reviewers are highlighted in *blue color*, our response is provided in black color, and the text that was modified or added to the manuscript is marked with *red color*.

Response to the comments from Reviewer #1

Comment: *The authors have clearly disagree with my previous comments and they have not been addressed - the authors have not really made an attempt to understand them, even when they parallel the comments of other referees. The responses to my comments are not endearing and are not really appropriate - particularly the discussion of "what is a crystal" to which the response is - we have published a mini-review elsewhere (which contains significant opinion) and the conclusions of our other opinion are that we are correct!*

Response to the comments: We thank the reviewer for their additional comments, and for the very careful reading of the revised manuscript. As authors of this manuscript, we would like to reassure the reviewer that we take each of their comments (as well as those from other reviewers) with utmost respect and very seriously, and as in the first revision, in this second revision we have also tried our best to address their newly expressed concerns to the best of our ability. As authors (and reviewers or many other authors' manuscripts) we truly appreciate and value the time that each reviewer has taken to re-read the revised manuscript. As always, we very carefully read, discuss, try to understand, and we work to address every single comment from every reviewer, and we hope that in the revised version of this manuscript we have made the necessary changes to the reviewer's satisfaction. We are very happy to accept sensible and substantiated criticisms based on actual results and the respective conclusions, because that discussion always contributes to improvement of the manuscript's quality.

The article to which the reviewer refers in their comment discusses an issue that was indeed raised in one of the reviewer's comments (that is, whether the samples we have used are single crystals, I quote: "*The biggest flaw and most significant is a requirement that the authors demonstrate that these are single crystals - indeed their microfocus diffraction experiments show that they are not single and probably not crystals*"), and we thought that instead of elaborating the source of confusion around the terminology and its proper use in great details, a reference to this recently published minireview will help clarify some of the concerns regarding the crystallinity of the samples. In our response to

the reviewer's comments we have provided both a written explanation and experimental evidence (diffraction images) regarding the crystallinity of our sample, and we have further supported our explanation with this recent reference which provides additional information based not only on our results but also results from other authors, which we think saves space and time with our response. We do strongly believe that this recent peer-review greatly clarifies the terminology and the proper and adequate use of both terms "single crystal" and "crystalline". However, if the reviewer thinks otherwise, we are happy to provide more detailed explanation in our response to the reviewer's comments and in support of our claims.

Comment: *Having noted and read the extensive, but too long and unconvincing responses to the referees I have reread the paper fresh to consider it in the absence of my previous comments - ie as if it were a fresh submission.*

Response to the comment: We again thank the reviewer for their time and attention with the careful reading of the manuscript and the comments. Our extensive comments are aimed at better explaining the changes we have made to respond to the reviewer's comments, and we hope the reviewer appreciates the effort we have taken to address their new comments, which were not part of their first set of comments.

Comment: *1. Line 52 - this is statement is incorrect. Plastic bending results in the loss of discrete diffraction - see Thomas et al.*

Response to the comment: We thank the reviewer for expressing their concern regarding the given statement. We are well aware of the work by Thomas, Spackman et al., and we have now modified the statement, and we have also included the relevant references. The following text was added in the main text of the revised manuscript together with additional references.

"Upon elastic or plastic bending of crystals their diffraction ability is generally retained, as concluded from their discrete diffraction patterns.^{8,42-46} However in some cases loss of the discrete diffraction has been observed upon plastic bending.⁴⁷⁻⁴⁹"

References added in the reference list:

45. Owczarek, M. et al. Flexible ferroelectric organic crystals. Nat. Commun. 7, 13108 (2016).

46. Saini, A. K., Natarajan, K., & Mobin, S. M. A new multitalented azine ligand: elastic bending, single-crystal-to-single-crystal transformation and a fluorescence turn-on Al(III) sensor. Chem. Commun. 53, 9870-9873 (2017).

47. Chou, C-M., Nobusue, S., Saito, S., Inoue, D., Hashizume, D., & Yamaguchi, S. Highly bent crystals formed by restrained π -stacked columns connected via alkylene linkers with variable conformations. *Chem. Sci.* 6, 2354–2359 (2015).

48. Liu, H., Bian, Z., Cheng, Q., Lan, L., Wang, Y., & Zhang, H. Controllably realizing elastic/plastic bending based on a room-temperature phosphorescent waveguiding organic crystal. *Chem. Sci.* 10, 227–232 (2019).

49. Rajca, A. et al. Functionalized Thiophene-Based [7]Helicene: Chiroptical Properties versus Electron Delocalization. *J. Org. Chem.* 74, 7504–7513 (2009).

Comment: 2. Line 64-68 the authors assert a new mechanism of bending. This is not supported - it appears that the mechanism of bending results from slippage of molecules (which is accompanied by a phase change), not a new mechanism. This is also a chicken and egg question.

Response to the comment: We are not sure from the comment whether the reviewer in their comment is referring to the mechanism of bending or the mechanism of phase transition. The *slippage* of molecular hydrogen-bonded tapes in this phase transition is the mechanism of the *phase transition* which causes *bending*, so the two processes are related to each other. We provide further clarification and more elaborate answer below.

We would like to reiterate here that as the main concept in this work we have proposed and provided experimental evidence (obtained by X-ray diffraction as well as by using other methods) of a distinctly new mechanism for plastic bending of molecular crystals that occurs by mechanically induced phase transition.

Therefore, the mechanism of the bending is mechanically induced phase transition. An illustration of this mechanism is provided in Figure 4D. This is entirely new mechanism for plastic bending of molecular crystals, where in the bent crystal two phases co-exist across a phase boundary, and the bending occurs irrespective of delamination and sliding of molecular slabs, a mechanism which is well established and common for plastic bending of some other plastically bendable crystals. As a support of this distinct mechanism, we have also demonstrated that the bent crystals of TA undergo a shape-memory effect, which is based on the reverse phase transition. This effect would not have been possible if the bending occurred simply by delamination and sliding of slabs.

Specifically, application of localized pressure in form II of TA induces a phase transition on the concave side of the crystal, and the two phases coexist in the bent region, as we have confirmed by X-ray diffraction analysis. We would like to note that the material that we report here provides the first ever example of plastic bending by a mechanically induced phase transition, and the form II of TA is the first compound which shows phase transition in the bent region upon application of force. We believe that similar bending is possible with other biphasic materials. The relatively soft nature of form I and the small structural difference between the two phases stabilizes the resulting bimorph in the bent region at room temperature. The bimorph can be converted to a single phase by heating

or cooling, and this results in apparent partial recovery of the straight shape of the crystal, which visually appears as a shape-memory effect.

Having that said, the mechanism of the phase transition, or the change between the two phases, which can be induced by either heating/cooling, occurs via slippage of molecular chains in a single-crystal-to-single-crystal manner. This means that the main difference between the two phases is offset of the molecular tapes from each other, which results in slight difference in the hydrogen bonds, as is clearly explained in the text. During the phase transition the molecular tapes of hydrogen-bonded TA molecules slide in respect to each other and take up new positions relative to each other, which results in change in the centroid-to-centroid distances. In the current work, the mechanism for plastic bending of the crystal by a mechanically induced phase transition and formation of a bimorph in the bent region is unique, although the phase transition does proceed through slippage of molecular tapes of hydrogen-bonded molecules. We also confirm that TA form II is the first example which shows phase transition in the bent region upon mechanically induced bending, and exist with form II and form I in the convex and concave regions respectively in the bent part of the bent crystal.

Comment: *3. There are not statistics or errors or objective measures the restorative effects. It is likely that the origins of these effects are defects and/or phase boundaries introduced upon deformations*

Response to the comment: We note that it is generally very difficult to quantify the restorative effects, however, in the meantime, to respond to the reviewer's comments, we have performed additional experiments that further confirm our claims. The new results are provided in the revised version of the supplementary material. We also note that the defects and phase boundaries play very different role in this mechanism.

We would like to reiterate here that according to the mechanism that we have established, the shape restoration of form II TA crystals occurs after partial phase transition to form I. Upon mechanical bending, the side that is impacted of form II TA crystal transforms to form I and remains in that phase in the bent region. Therefore, both form II and form I coexist in the convex and the concave regions of the kink of the bent crystal, and can be converted to each other either by heating or cooling. This results in apparent partial recovery of the straight shape of the crystal, and is observed as the shape-memory effect.

The main reason for the bent shape is the formation of a bimorph, which necessarily includes a phase boundary between the two phases. Namely, the conversion of form II to form I in the bent region during mechanical bending leads the generation of phase boundary between the two phases. When the crystal is heated, one of the phases converts to the other phase, so that the phase boundary disappears and the crystal straightens. Therefore, the main driving force of the shape-memory effect is the formation of a phase boundary. If there was no phase boundary, and the bending was due only to formation of defects, the crystal would not be able to recover its shape. Having said that, defects are always created when a soft crystal is impacted mechanically. However, these mechanically created defects can not be restored thermally, that is, if the crystal was bent by delamination and sliding of slabs, its original shape will never be recovered by heating.

In order to respond to the reviewer's comments, we have now added the following figure in the supplementary material. This figure shows quantification, expressed as the angle of the bent crystal by heating and cooling of the three different crystals. These results clearly show the reversibility of the shape memory effect.

Supplementary Figure 25 | Shape memory effects of crystals of form II terephthalic acid. These bent crystals were taken over the phase transition to form I by heating from 290 K to 360 K and cooled to 290 K. Scale bars: (A) 600 μm, (B) 800 μm, (C) 500 μm, (D) 700 μm, (E) 900 μm.

Comment: 4. *There are many statements like "it develops cracks on its surface ... but the slabs remain conjoined" - there is no evidence for this other than a photo - how do the authors know the cracks are only on the surface!!!?*

Response to the comment: We thank the reviewer for this comment which might have been confusing in the way it was stated. The development of the cracks has been mentioned once in the main text and once in the caption of Figure 2. We have corrected the statement by removing the reference to the crystal surface in both cases. Also, we note that when mechanical pressure is applied on (001) face of form II TA crystal, cracks appear on the crystal. This has been confirmed by using optical microscope and scanning electron microscope. In order to respond to the reviewer's comments, an SEM image of the damaged crystal is now included in the Supplementary Information.

Supplementary Figure 24 | SEM image of a mechanically damaged form II crystal of TA.

Comment: 5. *In figure 2 a mechanism of "apparent" restoration is proposed - this suggests that there is no restoration, it just appears to be restored!*

Response to the comment: We agree with the reviewer that this formulation can be confusing. We have modified the caption of Figure 2 to remove the word "apparent".

Comment: 6. *The authors use microfocus diffraction to analyze the inside and outside of a bent crystal and determine that two phase are present. By definition the resulting biphasic material is no longer a crystal - but that is of little consequence. The authors are unable to face-index these positions and as the crystals are triclinic the settings are arbitrary. Therefore there can be no conclusion drawn about theta vs theta prime. The changes in molecules are important not the changes in unit cells.*

Response to the comment: We would like to not that in the bent region, form II is converted to form I upon mechanical bending of form II TA crystal. In the bent region of this bent crystal, form I and form II coexist in the concave and convex sides of the crystal, respectively. We have performed microfocus X-ray diffraction on the convex and concave parts and we determined the structures of the two phases. We have also performed face indexing of the bent region of the bent crystal. This proved to be extremely challenging because the limited section of the converted bent region has to be identified and separated before performing face indexing. However, we have now included these face indexing images in the Supplementary Information a Supplementary Figure 26. The mechanism of bending is proposed and discussed in the main text with respect to the changes in the relative molecule positions in both form I and form II.

Supplementary Figure 26 | Face indexing of partially converted form II crystal.

Comment: 7. *It is much more likely that the reason for bending is planes of molecules slipping past each other, as noted by the authors between lines 270-306. Indeed inspection of the two crystal structures of the two polymorphs shows that the main difference between the two is a slippage of 0.7 Å along a chain in polymorph 2 compared to polymorph 1. All other distances (ie separations between chains etc) remains identical. This slippage is has been observed by the authors by their experiments. This slippage will be responsible for the bending - slippage is the mechanism - this is a known mechanism, not a new mechanism.*

Response to the comment: We would like to refer here to the above comment 2 of the reviewer, where this has been already pointed out, and also to our response to this-as the reviewer has posed and qualified it—“chicken and egg question”.

What we believe is being confused here is the mechanism of the bending and the mechanism of the phase transition, which are related to each other. The mechanism of the phase transition is the slippage of molecular layers. The mechanism of the bending is that very same phase transition. We think that these two can not be separated because they are related. The slippage of molecules has been, as pointed out by the reviewer, described as a mechanism of the thermally induced phase transition between one and another form of the same unbent crystal. However, what we propose here is that this phase transition occurs when the crystal is pressed and bent. In such case, the mechanism of bending is the mechanically induced phase transition in only one section of the crystal. As a result, a bilayer is formed. This mechanism of bending is very different from the previously observed mechanism which is based on delamination.

To clarify this further, Figure 4C describes the mechanism of plastic bending by delamination. Plastic bending usually occurs by partial delamination along the crystal length whereby slabs of the crystal slide atop each other but remained bound to each other. In effect, the interfacial stress exerted during bending induces small expansion of the individual layers, although the deformation of each layer may still be detected. The original shape of such bent crystal can never be recovered by heating. If the crystal is relatively short, the angles between the faces of the crystal at its termini may change; if the crystal is long, the delamination is not normally expected to proceed throughout the entire crystal (for example, see Panda, M. K. *et al. Nat. Chem.* 2015; Saha, S. *et al. Acc. Chem. Res.* 2018).

We would like to bring to reviewer attention that in the current work we have proposed a new mechanism for plastic bending by phase transition, as explained in Figure 4D. This is an entirely new mechanism for plastic bending and irrespective of delamination and slippage of slabs. In this case, application of localized stress induces a partial phase transition on the concave side of the crystal, and the two phases coexist in the bent region. The first ever example for plastic bending by phase transition is studied in this work and form II TA is the first compound which shows phase transition in the bent region upon mechanical bending.

Comment: *8. The authors haven't discussed the similar results of terephthalamide - a very similar molecule, with similar crystal packing. In this case application of stress results in a phase change and bending. Removal of the force results in the reversal of the phase change and restoration of the original shape. The only potential differences are: 1. lack of hysteresis 2. elastic vs plastic behaviour. The authors need to discuss this in detail and propose why one is elastic and one is plastic for there to be any real significant value here. It is disingenuous to ignore this previous careful study.*

Response to the comment: We have already cited this article in the previous versions of our manuscript. In the revised manuscript, we have now included a paragraph with comparison between terephthalamide and terephthalic acid:

“Takamizawa and Miyamoto reported superelasticity in terephthalamide (TPA) crystal which is structurally similar to terephthalic acid (TA).³³ Even that both terephthalic acid

and terephthalamide shows shape memory behavior they also show distinct properties. The α -phase of the terephthalamide transform to β -phase when pressure is applied by using a metal blade. The transition is reversible upon releasing the pressure. TA, on the other hand, shows a reversible phase transition upon heating and cooling apart from the bimorphic behavior in the bent region. Similar to TA, molecules in TPA associate to form sheets by hydrogen bonds and each terephthalamide molecule has four sites to form hydrogen bonds. By applying shear stress on TPA at room temperature, the uniform -A-A-A-A- sheet arrangement is transformed into an alternative and unstable -A'-B-A'-B- arrangement. The $\pi\cdots\pi$ stacking distance in A' is reduced to 3.209 Å from 3.500 Å and increased to 3.900 Å from 3.500 Å in B. The rotation of the molecular tapes observed in B- arrangement. This structural instability might result in elastic behavior. In TA during mechanical bending the molecular chains slide and relocate to new positions. During this transformation, $\pi\cdots\pi$ stacking distance changed to 3.665 Å from 4.973 Å. These changes eventually result in plastic behavior.”

Reference: (33) Takamizawa, S. & Miyamoto, Y. Superelastic Organic Crystals. *Angew. Chem. Int. Ed.* **53**, 6970–6973 (2014).

Comment: *At present I don't believe this paper is suitable for publication, because of the low level of scholarship/overhyped claims. If it were reworked I think it would be a valuable contribution.*

Response to the comment: We would like to confirm that we have tried to respond to all suggested changes. The originality of the work presented in this manuscript rests on the proposed new mechanism for plastic bending of a molecular crystal by a phase transition. TA form II crystal is the first example which shows bimorphs in the bent region of a mechanically bent crystal. We have characterized the observed properties with multiple analytical techniques, provided direct evidence of the mechanism, and we have provided a discussion of the results in the manuscript. We hope to have addressed the reviewer's concerns in a satisfactory manner.

Response to the comments from Reviewer #2

Comment: *I carefully read the responses from the authors to my comments/suggestions as well as those raised from other reviewers. In my opinion, the authors did an amazing job and fully addressed my concerns and incorporated my suggestions, including challenging mechanical characterization, the paper is much more refined now, a general public appeal is much clearer and I recommend accepting it for publication at Nature Comm in the present form.*

Response to the comment: We thank the reviewer for the generally positive assessment of the contents of our manuscript. We also appreciate that the reviewer recognizes the relevance and the possible impact of the results presented in this manuscript for the field of actuating materials.

Response to the comments from Reviewer #3

Comment: *The authors have addressed my concerns in the revision, and I support publication of the manuscript in its current form.*

Response to the comment: We thank the reviewer for the generally positive assessment of our work.

Response to the comments from Reviewer #4

We thank the reviewer for the careful reading of the revised manuscript and our detailed response, the additional comments, and the suggested changes. All comments have been addressed, as it is explained below.

Comment: *The authors have made significant effort to thoroughly address the concerns from the manuscript.*

Response to the comment: We thank the reviewer for the generally positive assessment of the work.

Comment: *Two questions that still arise from the response are regarding these responses:*

1) Original Comment: It would be interesting to see if there is a critical strain for forming I on the compressed concave side of the crystal. With less than critical strain, then both sides of the crystal even after bending may still be Form II.

Response to the comment: In order to verify this, we bent several crystals using three-point bending method and the critical strain was found to be $2.5 \pm 0.2\%$. We did not observe any phase transition below the critical strain.

New comment: For Figure S22, is the plateau that is observed definitively related to the Form II to I transition? Is it possible that it can be from gliding of the planes?

Response to the comment. The conclusion of the observed plateau as a result of phase transition is based on comparison of the stress-strain curves of TA with those of other

systems which show plastic deformation due to delamination and sliding of molecular slabs or layers (such as hexachlorobenzene). While defects are always created when a crystal is mechanically impacted, the restoration of the crystal's shape upon heating conforms that such mechanically generated defects are not the main contributor to the bending. Crystals that are bent by delamination do not recover their shape upon heating until they melt or sublime (as an illustration, we would like to bring to the reviewer's attention the supplementary movie 5 in the following reference, which shows melting of a bent crystal of hexachlorobenzene without any shape recovery: Panda, M. K. et al. Nat. Chem. 7, 65–72 (2015)).

Comment: 2) *Original Comment: Also authors mentioned in the same paragraph that: "The slight mismatch of the two unit cells, which—similar to their coexistence as alternating layers in a straight partially converted crystal—remain attached at the phase boundary, stabilizes the bimorph and the bent shape of the crystal is retained (Figure 4D)."*

As authors show the lattice structure in Figure 5A the mismatch at the phase boundary looks significant (for instance around 26.3 % decrease in a-axis length); which do not mean slight mismatch of the two unit cells. And in Figure 5B on the right schematic exaggerated the lattice dimension of Phase I to seem like it matches to that of Phase II.

Response to the comment: We have now changed the sentence accordingly in the revised version and also removed the exaggerated lattice dimension of phase I in Figure 5B

New Comment: Our concern of the mismatch of the two unit cells and especially the 26.3% difference in the a axis is still not well addressed. Specifically, in figure 5B, even in the revised figure, it does not accurately show the 26% difference in the a axis, which is significant. It seems unlikely that they would be sharing a common phase boundary.

Response to the comment: We agree with the reviewer that in the original and revised versions of Figure 5 the schematics did not faithfully reflect the relative size of the two unit cells (the difference of 26.3% in the length of the a axis is indeed quite significant). We have now corrected Figure 5 (corrected panel a1 and panel b2, and simplified panel c2) to indicate better the relative ratio of the length of the unit cell axes and the respective angles between the two phases. We have also double-checked the unit cell orientations in the bent crystal and we confirmed that the two lattices in the bimorph are conjoined at the unit cell's a axes. Since we could not observe any physical separation by delamination upon bending, the retention of the macroscopic integrity of the crystal which was evident in all our experiments may be due to partial relaxation of the strain imposed by the curvature of the crystal that is generated after bending. The fact that the new form generated on the concave side of the crystal (phase I) has a shorter a axis than the original form (phase II), and not vice versa, goes along those lines. In the main text, we have modified the statements to highlight this difference between the two unit cells:

"The retention of crystal integrity may be attributed to the identical crystal symmetry of the two phases (triclinic, space group P-1) and the marginal difference in their unit cell

volumes of only 0.07% although there is a significant difference in the individual unit cell axes (Supplementary Table 4).”

The revised Figure 5 is provided below for convenience:

Figure 5 | Mechanism of the phase transition during bending and shape-memory effect of a single crystal of terephthalic acid. (A) Transformation of the crystal between forms II and I, and relationship between the unit cells in the two phases. (a1) Transformation between the unit cell orientations of form II and form I viewed normal to the crystallographic faces (001) and (010). (a2) Images of the crystal in the two phases with face indices. (a3) Variation in the dimensions and shape of the unit cell, shown as different views. (B) Mechanism of transformation of the two-dimensional crystal lattice (cartoon) during the phase transition induced by application of local pressure. (b1) Crystal of form II and schematic of its lattice before bending. The straight crystal of form II can be

bent by applying pressure on its (010) or (0 $\bar{1}$ 0) face. The lattice here is shown along the crystallographic [100] direction. (b2) Crystal and schematic of its lattice after bending. The bent crystal is a bimorph with two coexistent phases in the bent region separated by a phase boundary. (C) Mechanism of the shape-restorative effect. (c1) A bimorph in the kink of the bent crystal, shown with its lattice and unit cell. Unit cell orientations of both form II and form I in the bent region. (c2) Partial restoration of the straight shape of the crystal, and schematic representation of the underlying phase transition to form I.

Reviewers' comments:

Reviewer #1 (Remarks to the Author):

On the basis of the referee's comments the authors have made some changes to the paper which substantially improve it, however, more work needs to be done. As I have said in my previous two reports this paper goes well beyond what is proven and into the realm of conjecture - little attempt has been made to tone down these inappropriate (and probably misleading statements). I continue to think that the results are interesting and worthy of publication in nature communications if the paper is further modified. As my comments, particularly about what is new, and known, and what is not continue to fall on deaf ears, here instead I have listed, by line the statements that need to be changed and/or removed.

1. Line 2 - the title. Completely inappropriate - this paper really isn't about shape-memory effects in general - it is about bending TA crystals.

2. Line 16-18. The mechanism is slippage. This is not new - slippage is different to delamination. It is interesting that this material when bent, has two different phases, but that doesn't mean it is a different mechanism. I completely disagree with the response to my previous comments in this regard.

3. Line 32 - grammar needs revision

4. Line 52-53. This is incorrect. Plastic deformation generally results in the loss of discrete diffraction.

5. Lines 55-61. This also needs serious revision given 4 - this displays a misunderstanding of the theory of diffraction.

6. Line 135 - no evidence is presented of the lack of delamination - the authors should reword to say "no visible delamination"

7. Line 145 "complete" recovery? By what measure?

8. Line 229-231: there is no evidence for this - if this is hypothesis it should state so.

9. Line 232-233 - as discussed in lines 347-361, this is not the first example of this.

10. Line 234-235 - this is incorrect. The mechanism is still slippage, just on the molecular level, not visible delamination of the crystal.

11. Lines 270-286. This needs reworking. The mechanism of plastic deformation is not delamination it is slippage. Correct and/or remove.

12. Lines 288-289. This is incorrect and needs to be removed.

13. Lines 335-337. There is not evidence of rotation presented - this is incorrectly assumed based on the misunderstanding of the cell relationships. You need to consider the orientation matrices not just the cell axes if you want to find out what is happening. Alternatively consider the molecules not the cell axes.

14. Lines 359-360. This statement is correct, but inconsistent with the statements above. ie the mechanism of bending is slippage.

The paper will only be acceptable in my view if all of the above are correct. I note here I have not suggested similar changes to the figure captions, but similar work will need to be done.

Response to the comments from the reviewers

We have considered all comments from the reviewer, and we tried to address them to the best of our ability. Unless stated otherwise, all numbers of the figures and supplementary materials refer to the *revised* version of the manuscript. Together with this submission we have provided marked copies of the main text and the supplementary materials where the changes to the original version have been marked. For convenience, in what follows, the original comments from the reviewers are highlighted in *blue color*, our response is provided in black color, and the text that was modified or added to the manuscript is marked with *red color*.

Response to the comments from Reviewer #1

Comment: *On the basis of the referee's comments the authors have made some changes to the paper which substantially improve it, however, more work needs to be done. As I have said in my previous two reports this paper goes well beyond what is proven and into the realm of conjecture - little attempt has been made to tone down these inappropriate (and probably misleading statements). I continue to think that the results are interesting and worthy of publication in nature communications if the paper is further modified. As my comments, particularly about what is new, and known, and what is not continue to fall on deaf ears, here instead I have listed, by line the statements that need to be changed and/or removed.*

Response to the comment: As in the previous round of revisions, we have addressed each individual comment from the reviewer. The changes made in the manuscript are available from the marked version of the manuscript. A point-by-point response to the comments and the changes made is provided below.

Comment: *1. Line 2 - the title. Completely inappropriate - this paper really isn't about shape-memory effects in general - it is about bending TA crystals.*

Response to the comment: We have considered this suggestion, and decided to retain the title of the manuscript. What we report in this manuscript is a phenomenon whereby a crystal which has been bent mechanically, recovers its shape (partially or completely) after it is taken over a phase transition. This phenomenon is analogous to that observed in shape-memory polymers, and the main subject in this manuscript is that the observations on the shape change of the material are similar to those observed for shape-memory polymers.

Comment: *2. Line 16-18. The mechanism is slippage. This is not new - slippage is different to delamination. It is interesting that this material when bent, has two different*

phases, but that doesn't mean it is a different mechanism. I completely disagree with the response to my previous comments in this regard.

Response to the comment: Indeed the molecular sliding (“slippage”) is the mechanism of the phase transition which occurs during the mechanical bending, and it is different from delamination. We have added the “slippage mechanism” to the description in the abstract. The reformulated description of the process in the abstract is:

“Here we report that organic crystals can also be bent while they undergo a mechanically induced phase transition by slippage of molecules and without delamination, whereby the overall crystal integrity is retained.”.

Comment: *3. Line 32 - grammar needs revision*

Response to the comment: The sentence was changed to:

“Dynamic molecular crystals are mechanically compliant materials which are thought to combine properties of soft matter such as polymers, and hard matter such as inorganic materials.”

Comment: *4. Line 52-53. This is incorrect. Plastic deformation generally results in the loss of discrete diffraction.*

Response to the comment: We have modified the sentence as it follows:

“Upon elastic bending of crystals their diffraction ability is generally retained, as concluded from their discrete diffraction patterns.^{8,42–46} In some cases loss of the discrete diffraction has been observed upon plastic bending.^{47–49}”

Comment: *5. Lines 55-61. This also needs serious revision given 4 - this displays a misunderstanding of the theory of diffraction.*

Response to the comment: We have modified the text accordingly:

“The structure determination of bent crystals by using conventional diffraction analysis is not trivial⁴⁴ due to the existing range of unit cell orientations and increased defects in the bent section that normally result in streaky diffraction profiles.”

Comment: *6. Line 135 - no evidence is presented of the lack of delamination - the authors should reword to say "no visible delamination"*

Response to the comment: The sentence was reworded accordingly:

“However, unlike other examples of plastically bendable crystals³⁸ the bending of TA crystals occurs without visible delamination.”

Comment: 7. Line 145 "complete" recovery? By what measure?

Response to the comment: We inspect and determine the recovery based on the curvature of the crystal. However, we acknowledge that even the shape has been recovered, there may be residual defects in the crystal. To clarify this, and to avoid misunderstanding the sentence was reformulated to remove the word “complete” and the method which was used to detect the recovery was added. In the text and in the Supplementary Materials (Supplementary Movies 8 and 9) we have provided evidence of the shape recovery of the crystals. The modified sentence is copied below:

“In some cases, we were astonished to observe recovery of the straight shape of the crystal, based on its curvature (Supplementary Movies 8 and 9).”

Comment: 8. Line 229-231: there is no evidence for this - if this is hypothesis it should state so.

Response to the comment: The sentence was reformulated to:

“We hypothesize that the retention of crystal integrity may be attributed to the identical crystal symmetry of the two phases (triclinic, space group $P\bar{1}$) and the marginal difference in their unit cell volumes of only 0.07% although there is a significant difference in the individual unit cell axes (Supplementary Table 4).”

Comment: 9. Line 232-233 - as discussed in lines 347-361, this is not the first example of this.

Response to the comment: The sentence was reformulated to:

“The results present evidence of a phase transition induced by bending in a crystal that has a metastable phase.”

Comment: 10. Line 234-235 - this is incorrect. The mechanism is still slippage, just on the molecular level, not visible delamination of the crystal.

Response to the comment: The sentence was removed.

Comment: 11. Lines 270-286. This needs reworking. The mechanism of plastic deformation is not delamination it is slippage. Correct and/or remove.

Response to the comment: We note that the statements in the sentences 270-286 are based on literature data (not on results presented in this article), and the appropriate literature references are cited. This paragraph discusses the general mechanisms of elastic and plastic bending that have been well established and are not a result of this work. We are unsure how this should be reworked to make it more understandable, but we have removed the word “delamination” if that is what caused the misunderstanding:

“On the other hand, the plastic bending is regularly accompanied by sliding of molecular layers, whereby the individual layers glide but remain in contact with each other.^{35–39,52} This is often observed as evolution of striations on the crystal surface perpendicular to the plane of bending (Figure 4C; Supplementary Note 2).”

Comment: *12. Lines 288-289. This is incorrect and needs to be removed.*

Response to the comment: The sentence was removed.

Comment: *13. Lines 335-337. There is not evidence of rotation presented - this is incorrectly assumed based on the misunderstanding of the cell relationships. You need to consider the orientation matrices not just the cell axes if you want to find out what is happening. Alternatively consider the molecules not the cell axes.*

Response to the comment: We have modified the corresponding sentence by removing the reference to the crystallographic axis, which may have caused the confusion. The revised sentence is copied below:

“This causes sliding and rotation of the molecular tapes in form II to nearly 30° on the concave side, where they reform the sheets along the crystallographic c axis, and the concave part of the crystal is transformed to form I.”

To respond to this comment and to clarify the mechanism further, we have also modified Figure 5 which now shows individual molecules in panels b1 and b2. The caption of Figure 5 was revised accordingly. The orientation of the molecules shown in the revised figure is based on the face indices and the crystal structures of the two phases in the bent region determined by X-ray diffraction analysis.

We would like to point out here that unlike twinned structures, where both phases are identical but have different orientations of their unit cells, in our case the two phases on the opposite sides of the crystal are structurally different (based on the evidence provided by X-ray direction), and therefore their relationship can not be explained by using orientation matrices. Indeed, our structure determination of the two phases has provided the face indices and the unit cell relation between the phases across the phase boundary. The ‘rotation’ to which we refer in this context pertains to the interplanar angle between the molecules in the two phases. We hope that the new figure explains this relation more clearly. In order to facilitate the explanation further, we have included an additional figure

in the supplementary material (Supplementary Figure 27), and we have added reference to this figure in the caption of Figure 5. This figure shows a zoomed-in representation of the two structures in the bent region and the relationship between the molecular orientation in the respective structures.

Comment: *14. Lines 359-360. This statement is correct, but inconsistent with the statements above. ie the mechanism of bending is slippage.*

Response to the comment: The statement was retained. The other statements were modified.

Comment: *The paper will only be acceptable in my view if all of the above are correct. I note here I have not suggested similar changes to the figure captions, but similar work will need to be done.*

Response to the comment: The suggested changes were introduced in the revised text. We hope that the revised text is acceptable in its current form. Figure 5 was also modified to better reflect the mechanism of the phase transition.

REVIEWERS' COMMENTS:

Reviewer #1 (Remarks to the Author):

My previous comments were absolutes. I would reject this manuscript as the title has not been changed. The other responses and adjustments while not going as far as I would like, are now acceptable.

Why has this been like pulling teeth? There really isn't any need for that.

Reviewer #2 (Remarks to the Author):

I was among the original reviewer's and during last iteration i already was in favor of accepting the manuscript for publication to Nature Comm. This has not changed. I carefully looked at each individual question and corresponding author to the first reviewer and in my opinion authors to the best of their ability addressed all the concerns. Based on this and after again carefully going over the revised manuscript, I maintain the paper should be accepted in its present work and provides an important contribution to the field as well as generate more awareness of the solid state transformations in molecular crystals.

Alexei Tivanski
Associate Professor of Chemistry
Department of Chemistry
University of Iowa

Response to the comments from the reviewers

REVIEWERS' COMMENTS:

Reviewer #1 (Remarks to the Author):

Comment: *My previous comments were absolutes. I would reject this manuscript as the title has not been changed. The other responses and adjustments while not going as far as I would like, are now acceptable.*

Why has this been like pulling teeth? There really isn't any need for that.

Response: The title of the manuscript was modified to the extent that it does not alleviate the impact of the results, while it accurately reflects the contents of the material presented in the manuscript. All other comments have been addressed.

Reviewer #2 (Remarks to the Author):

Comment: *I was among the original reviewer's and during last iteration i already was in favor of accepting the manuscript for publication to Nature Comm. This has not changed. I carefully looked at each individual question and corresponding author to the first reviewer and in my opinion authors to the best of their ability addressed all the concerns. Based on this and after again carefully going over the revised manuscript, I maintain the paper should be accepted in its present work and provides an important contribution to the field as well as generate more awareness of the solid state transformations in molecular crystals.*

*Alexei Tivanski
Associate Professor of Chemistry
Department of Chemistry
University of Iowa*

Response: The authors of this manuscript thank Dr. Tivanski for the time he took to read the manuscript and the useful comments he provided that contributed for its further improvement.